# ELIMINATING INDUCTIVE BIAS IN REWARD MODELS WITH INFORMATION-THEORETIC GUIDANCE

**Zhuo Li**[1,2,*]**, Pengyu Cheng**[1,†]**, Zhechao Yu**[1]**, Feifei Tong**[1]**, Anningzhe Gao**[2]**,**
**Tsung-Hui Chang**[2]**, Xiang Wan**[2,†]**, Erchao Zhao**[1]**, Xiaoxi Jiang**[1]**, Guanjun Jiang**[1]

[1]Qwen Large Model Application Team, Alibaba
[2]Shenzhen Research Institute of Big Data, The Chinese University of Hong Kong (Shenzhen)
[†]Corresponding authors.

## ABSTRACT

Reward models (RMs) are essential in reinforcement learning from human feedback (RLHF) to align large language models (LLMs) with human values. However, RM training data is commonly recognized as low-quality, containing inductive biases that can easily lead to overfitting and reward hacking. For example, more detailed and comprehensive responses are usually human-preferred but with more words, leading response length to become one of the inevitable inductive biases. A limited number of prior RM debiasing approaches either target a single specific type of bias or model the problem with only simple linear correlations, *e.g.*, Pearson coefficients. To mitigate more complex and diverse inductive biases in reward modeling, we introduce a novel information-theoretic debiasing method called **D**ebiasing via **I**nformation optimization for **R**M (DIR). Inspired by the information bottleneck (IB), we maximize the mutual information (MI) between RM scores and human preference pairs, while minimizing the MI between RM outputs and biased attributes of preference inputs. With theoretical justification from information theory, DIR can handle more sophisticated types of biases with non-linear correlations, broadly extending the real-world application scenarios for RM debiasing methods. In experiments, we verify the effectiveness of DIR with three types of inductive biases: *response length*, *sycophancy*, and *format*. We discover that DIR not only effectively mitigates target inductive biases but also enhances RLHF performance across diverse benchmarks, yielding better generalization abilities. The code and training recipes are available at `https://github.com/Qwen-Applications/DIR`.

## 1 INTRODUCTION

Aligning Large Language Models (LLMs) (OpenAI, 2024; Touvron et al., 2023; Yang et al., 2024) with human values is a fundamental technique to guarantee the helpfulness and harmlessness of LLM responses, which has been widely applied in various open-domain scenarios (Ouyang et al., 2022b; Kimi et al., 2025; Touvron et al., 2023; Gemini, 2025; OpenAI, 2024; Hu et al., 2025; Dai et al., 2025). Toward more human-preferred LLM behaviors, reinforcement learning from human feedback (RLHF) (Ouyang et al., 2022b; Rafailov et al., 2024b; DeepSeek-AI, 2025) has become the mainstream approach, which first trains a reward model (RM) on a collection of human preference response pairs, then scores the LLM's responses with the learned RM as the rewards to conduct a reinforcement learning (RL) (Ouyang et al., 2022b). Although dominantly applied in LLM post-training (DeepSeek-AI, 2025; Touvron et al., 2023), RLHF has continuously been criticized by its training instability, which can easily lead to LLM's training collapse and overfitting (Rafailov et al., 2024b; Zhu et al., 2024; Yu et al., 2025).

Among the many factors leading to the training instability of RLHF, the issue of reward model hacking is non-negligible: due to the low quality of human preferences (Zeng et al., 2024; Liu et al., 2025; Wang et al., 2025b; Liu et al., 2024a) that contains massive preference conflicts and inductive

---
[*]Work done during an internship at Alibaba.

biases, the reward model can easily be mislead by irrelative data attributes instead of targeting on the real content quality (Skalse et al., 2025; Gao et al., 2023; Amodei et al., 2016; Du et al., 2025; Li et al., 2025a). For example, annotators are always instructed to choose more informative responses, whereas more detailed responses usually have longer response lengths. Learning on this biased human feedback dataset can lead the reward model to ignore the true response quality and only favor responses with longer lengths (Singhal et al., 2023). Besides the response length bias, stylistic and format patterns (Zhang et al., 2025), and sycophantic phrasing (Sharma et al., 2023; Denison et al., 2024) have also been recently recognized as the typical *inductive biases* in reward modeling, which are unrelated to response quality but strongly correlated with the preference annotations (Sharma et al., 2023; Liu et al., 2024c). Learning with inductive biases can easily disrupt RMs' learning targets and critically hinders the reliability and the generalization ability of RLHF (Gao et al., 2023; Coste et al., 2023).

To mitigate inductive biases in reward modeling, a limited number of recent studies have made some preliminary explorations. Bu et al. (2025); Chen et al. (2024); Zhang et al. (2025) consider Pearson Coefficient (Benesty et al., 2009) as the bias measurement, which is then jointly minimized with the reward modeling loss. However, the Pearson Coefficient only captures the simplest *linear* correlation between RM and the bias attributes, which are not sufficiently applicable in more general scenarios. Shen et al. (2023) adds another RM head to predict response length score, which is only applicable with scalar types of inductive biases and lacks theoretical justification. Wang et al. (2025a) introduces overly restrictive external constraints, such as enforcing distributional invariance by minizmizing the Maximum Mean Discrepancy (MMD) (Gretton et al., 2012) between the chosen and rejected distributions. This approach risks distorting the reward landscape by inducing a collapse in the scores of functionally distinct response groups. In contrast, methods utilizing general compression strategies, such as the Information Bottleneck (Tishby et al., 2000) used in InfoRM (Miao et al., 2024), cannot guarantee the mitigation of inductive biases, since no explicit constraint is applied to the biased attributes within the optimization.

To uniformly eliminate inductive biases from reward modeling with theoretical guarantees, we propose an *information-theoretic* debiasing framework called **D**ebiasing via **I**nformation optimization for **R**Ms (DIR). Inspired by the information bottleneck methods (Tishby et al., 2000), we model the complicated inductive biases between irrelevant attributes and human preferences by the concept of mutual information (MI) from the perspective of information theory (Kullback, 1997). The proposed debiasing method maximizes the mutual information between the content quality of response pairs and the ground-truth preference label, while simultaneously minimizing the mutual information between the RM's preference prediction and the irrelevant bias attributes. To tackle the intractable MI calculation (Poole et al., 2019), we estimated the above objective with two variational bounds: the Barber-Agakov (BA) lower bound (Barber & Agakov, 2004) for the MI maximization and the contrastive log-ratio upper bound (CLUB) (Cheng et al., 2020) for the MI minimization. To further extend the method's applicability, we design a comparative regularizer that operates on relative bias attributes between response pairs, rather than on the isolated attributes of individual responses. This modification allows DIR to generally handle diverse and complex types of biases without distorting the underlying reward landscape, yielding broader application scenarios. We conduct extensive experiments on both LLM capability benchmarks (*e.g.*, GSM8K (Cobbe et al., 2021), MMLU (Hendrycks et al., 2021), ArenaHard (Li et al., 2024), and MT-Bench (Zheng et al., 2023)) and reward model benchmarks (*e.g.*, RM-Bench (Liu et al., 2024c) and RewardBench (Lambert et al., 2024)), under multiple bias settings including response length, formatting style, and sycophancy. The numerical results demonstrate that the proposed DIR method consistently outperforms existing debiasing baselines, yielding more robust and reliable alignment of LLMs.

## 2 PRELIMINARY

**Reinforcement Learning from Human Feedback (RLHF)** has become one of the essential training processes to align LLMs with human values (Ouyang et al., 2022a). With a well-learned reward model (RM) $r_\phi(\boldsymbol{x}, \boldsymbol{y})$ scoring the degree of human preference of generated response $\boldsymbol{y} \in \mathcal{Y}$ given input prompt $\boldsymbol{x} \in \mathcal{X}$, RLHF optimizes the LLM policy $\pi_\theta(\boldsymbol{y}|\boldsymbol{x})$ with the follow objective:

$$\mathbb{E}_{\boldsymbol{x} \sim \mathcal{X}, \boldsymbol{y} \sim \pi_\theta(\cdot|\boldsymbol{x})}[r(\boldsymbol{x}, \boldsymbol{y}) - \beta \cdot \mathrm{KL}[\pi_\theta(\boldsymbol{y}|\boldsymbol{x})\|\pi_{\mathrm{ref}}(\boldsymbol{y}|\boldsymbol{x})]], \tag{1}$$

where $\pi_{\mathrm{ref}}(\boldsymbol{y}|\boldsymbol{x})$ is the initial model policy served as a reference, $\beta > 0$ controls the strength of a Kullback-Leibler (KL) divergence (Csiszár, 1975) between the reference model $\pi_{\mathrm{ref}}(\boldsymbol{y}|\boldsymbol{x})$ and the

learning policy $\pi_\theta(\boldsymbol{y}|\boldsymbol{x})$. To train LLMs with the above objective, Proximal Policy Optimization (PPO) (Schulman et al., 2017) has been recognized as the mainstream optimization approach (OpenAI, 2024; Bai et al., 2023; Rafailov et al., 2024b). Group Relative Policy Optimization (GRPO) (Shao et al., 2024) further removes the critic model in PPO and uses a simplified group-related advantage approximation instead, which has shown competitive performance with practically simpler infrastructures (DeepSeek-AI, 2025; Yang et al., 2025).

**Reward Modeling** targets learning the human preference distribution via a parameterized reward model (RM) $r_\phi : \mathcal{X} \times \mathcal{Y} \to \mathbb{R}$, where $r_\phi(\boldsymbol{x}, \boldsymbol{y})$ is the predicted reward score of the input prompt $\boldsymbol{x}$ and the corresponding response $\boldsymbol{y}$. For every input $\boldsymbol{x}$, given a pair of response $(\boldsymbol{y}, \bar{\boldsymbol{y}})$, we can calculate the "preference" by comparing the reward scores: if $r(\boldsymbol{x}, \boldsymbol{y}) > r(\boldsymbol{x}, \bar{\boldsymbol{y}})$, then $\boldsymbol{y}$ is predicted as a more "preferred" response than $\bar{\boldsymbol{y}}$ (denote as $\boldsymbol{y} \succ \bar{\boldsymbol{y}}$) and vice versa. We use a binary indicator $\mathbf{1}_{\boldsymbol{y} \succ \bar{\boldsymbol{y}}}$ to represent the event of "human preference": $\mathbf{1}_{\boldsymbol{y} \succ \bar{\boldsymbol{y}}} = 1$, if $\boldsymbol{y} \succ \bar{\boldsymbol{y}}$; and $\mathbf{1}_{\boldsymbol{y} \succ \bar{\boldsymbol{y}}} = 0$, if $\boldsymbol{y} \prec \bar{\boldsymbol{y}}$. Then the RM predicting preference $\mathbf{1}_{\boldsymbol{y} \succ \bar{\boldsymbol{y}}}$ can be regarded as drawing a conditional Bernoulli (Chen & Liu, 1997) random variable from:

$$q_\phi\Big(\mathbf{1}_{\boldsymbol{y} \succ \bar{\boldsymbol{y}}} = 1 \Big| \boldsymbol{x}, \boldsymbol{y}, \bar{\boldsymbol{y}}\Big) = \frac{\exp(r_\phi(\boldsymbol{x}, \boldsymbol{y}))}{\exp(r_\phi(\boldsymbol{x}, \boldsymbol{y})) + \exp(r_\phi(\boldsymbol{x}, \bar{\boldsymbol{y}}))} = \sigma\Big(r_\phi(\boldsymbol{x}, \boldsymbol{y}) - r_\phi(\boldsymbol{x}, \bar{\boldsymbol{y}})\Big), \quad (2)$$

where $\sigma(\cdot)$ is a Sigmoid function. Note that the ground-truth human preference distribution $p^*(\mathbf{1}_{\boldsymbol{y} \succ \bar{\boldsymbol{y}}}|\boldsymbol{x}, \boldsymbol{y}, \bar{\boldsymbol{y}})$ is unknown. To optimize the reward model, we instead maximize the log-likelihood of $q_\phi$ with a group of human preference data $\mathcal{D}_{\text{Pref}} = \{(\boldsymbol{x}_i, \boldsymbol{y}_i^w, \boldsymbol{y}_i^l)\}_{i=1}^N$:

$$\mathcal{L}_{\text{RM}}(\phi) = - \mathbb{E}_{\mathbf{1}_{\boldsymbol{y} \succ \bar{\boldsymbol{y}}} \sim p^*} \Big[ \log q_\phi\Big(\mathbf{1}_{\boldsymbol{y} \succ \bar{\boldsymbol{y}}} \Big| \boldsymbol{x}, \boldsymbol{y}, \bar{\boldsymbol{y}}\Big) \Big] \approx -\frac{1}{N} \sum_{i=1}^N \Big[ \log q_\phi(\boldsymbol{y}_i^w \succ \boldsymbol{y}_i^l | \boldsymbol{x}_i, \boldsymbol{y}_i^w, \boldsymbol{y}_i^l) \Big]$$

$$= -\frac{1}{N} \sum_{i=1}^N [\log \sigma(r_\phi(\boldsymbol{x}_i, \boldsymbol{y}_i^w) - r_\phi(\boldsymbol{x}_i, \boldsymbol{y}_i^l))], \qquad \text{(by equation 2)} \qquad (3)$$

where each $\boldsymbol{y}^w \succ \boldsymbol{y}^l$ is annotated by human judgment with respect to the response content quality. Equation 3 is commonly recognized as the Bradley-Terry ranking loss (Bradley & Terry, 1952).

**Information-theoretic Methods** optimize deep models from the perspective of information theory (Chen et al., 2016; Hjelm et al., 2019; Yuan et al., 2021; Cheng et al., 2021). The core methodology of information-theoretic methods is modeling the feed-forward process of neural networks as an information channel transmission, where the correlation between different neural embeddings is measured by mutual information (MI) as:

$$I(\boldsymbol{x}; \boldsymbol{y}) = \mathbb{E}_{p(\boldsymbol{x}, \boldsymbol{y})} \Big[ \log \frac{p(\boldsymbol{x}, \boldsymbol{y})}{p(\boldsymbol{x})p(\boldsymbol{y})} \Big] = \text{KL}\Big[ p(\boldsymbol{x}, \boldsymbol{y}) \| p(\boldsymbol{x})p(\boldsymbol{y}) \Big], \qquad (4)$$

where $p(\boldsymbol{x}, \boldsymbol{y})$ is the joint distribution, and $p(\boldsymbol{x})$ and $p(\boldsymbol{y})$ are the marginal distributions. Due to its general ability to capture arbitrary non-linear correlations, MI has achieved considerable success as a learning objective in various deep learning tasks (Chen et al., 2016; Belghazi et al., 2018; Hjelm et al., 2019). However, due to the intractable expectation *w.r.t.* $p(\boldsymbol{x}, \boldsymbol{y})$, the exact MI value in equation 4 is challenging to compute, especially when only samples from $p(\boldsymbol{x}, \boldsymbol{y})$ are provided. To address this, several approximation methods have been proposed to estimate MI from samples using tractable variational bounds (Oord et al., 2018; Cheng et al., 2020; Belghazi et al., 2021). Barber-Agakov (BA) bound (Barber & Agakov, 2004) provides a simple lower bound approximation of MI, by introducing a variational approximation $q_\theta(\boldsymbol{y}|\boldsymbol{x})$:

$$I(\boldsymbol{x}; \boldsymbol{y}) \geq \mathbb{E}_{p(\boldsymbol{x}, \boldsymbol{y})}[\log q_\theta(\boldsymbol{y}|\boldsymbol{x})] + H[p] =: I_{\text{BA}}(\boldsymbol{x}; \boldsymbol{y}), \qquad (5)$$

where $H[p]$ is the entropy of the ground-truth distribution $p(\boldsymbol{x}, \boldsymbol{y})$. Besides, Cheng et al. (2020) propose a variational contrastive log-ratio upper bound (CLUB) also utilizing the variational approximation $q_\theta(\boldsymbol{y}|\boldsymbol{x})$:

$$I(\boldsymbol{x}; \boldsymbol{y}) \leq \mathbb{E}_{p(\boldsymbol{x}, \boldsymbol{y})}[\log q_\theta(\boldsymbol{y}|\boldsymbol{x})] - \mathbb{E}_{p(\boldsymbol{x})p(\boldsymbol{y})}[\log q_\theta(\boldsymbol{y}|\boldsymbol{x})] =: I_{\text{CLUB}}(\boldsymbol{x}; \boldsymbol{y}). \qquad (6)$$

By minimizing equation 6, the amount of information between $\boldsymbol{x}$ and $\boldsymbol{y}$ can be effectively reduced. We provide the proof of BA bound and CLUB in Appendix B.1& B.2. A well-known application of

information-theoretic methods is the *information bottleneck* (IB) (Tishby et al., 2000), which aims to learn a compressed but informative representation $h$ of an input $x$ to the output $y$ as a trade-off between two MI terms:

$$\min_{h} I(x; h) - \lambda \cdot I(h; y), \tag{7}$$

where hyper-parameter $\lambda > 0$ controls the balance between compressing the input $x$ and retaining relevant information for the prediction $y$. IB has been recognized as a powerful tool for representation learning and widely applied to diverse deep learning scenarios (Saxe et al., 2019; Wan et al., 2021; Federici et al., 2020).

## 3 METHODOLOGY

We begin by revisiting reward modeling from a perspective of information theory. Given an input query $x \in \mathcal{X}$ and a pair of responses $y, \bar{y} \in \mathcal{Y}$, we denote $b$ as a concerned bias attribute with respect to $(x, y, \bar{y})$. Our debiasing target is to learn a reward model $r_\phi(x, y)$ that produces predictions $\mathbf{1}_{y \succ \bar{y}}$ highly correlated with the content quality of response pairs $(y, \bar{y})$ while eliminating any indication of the pre-defined bias attribute $b$. Motivated by the information bottleneck method in equation 7, we model the debiasing objective as maximizing the mutual information between the input response content and the RM preference prediction, while minimizing the mutual information between the RM prediction and the bias attribute:

$$\max_{\phi} \underbrace{I(\mathbf{1}_{y \succ \bar{y}}; x, y, \bar{y})}_{\text{Preference Term}} - \lambda \cdot \underbrace{I(\mathbf{1}_{y \succ \bar{y}}; b)}_{\text{Debiasing Term}}, \tag{8}$$

where $\lambda > 0$ is a hyper-parameter balancing the trade-off between preference learning and debiasing. Ideally, minimizing equation 8 should encourage the reward model $r_\phi$ to capture the true performance signal from the input triplet $(x, y, \bar{y})$, while decreasing the reliance on the bias attribute $b$.

However, directly optimizing the mutual information-based objective is computationally intractable due to the difficulty in estimating mutual information in high-dimensional spaces (Poole et al., 2019). Hence, we follow the prior works (Oord et al., 2018; Cheng et al., 2021) and utilize the variational mutual information bounds (as in equation 5 & 6) to estimate the preference term and debiasing term of equation 8, separately.

**Preference Term Estimation.** Instead of directly enlarging $I(\mathbf{1}_{y \succ \bar{y}}; x, y, \bar{y})$, we can maximize its lower bound approximation by applying the BA estimator as in equation 5:

$$I(\mathbf{1}_{y \succ y}; x, y, \bar{y}) \geq \mathbb{E}_{p^*(x, y, \bar{y}, \mathbf{1}_{y \succ \bar{y}})}[\log q_\phi(\mathbf{1}_{y \succ \bar{y}} | x, y, \bar{y})] + H[p^*], \tag{9}$$

where $p^*(x, y, \bar{y}, \mathbf{1}_{y \succ \bar{y}})$ is the ground-truth joint distribution of human preference training data, and $H[p^*]$ is the entropy of the data distribution $p^*$ as a constant to the learning parameters. By equation 3, the expectation term in the right-hand side of equation 9 is exactly the commonly used Bradley-Terry ranking loss (Bradley & Terry, 1952) of reward modeling (Azar et al., 2024; Cheng et al., 2024). Hence, minimizing the RM ranking loss actually maximizes the mutual information between the preference prediction $\mathbf{1}_{y \succ \bar{y}}$ and the input triplet $(x, y, \bar{y})$, encouraging the reward model $r_\phi$ to output a higher score to the preferred response $y$. Therefore, given a batch of preference data $\mathcal{D}_{\text{Pref}} = \{(x_i, y_i^w, y_i^l) | y_i^w \succ y_i^l\}_{i=1}^{B}$, we maximize the following RM ranking loss to approximate the preference term in equation 8 instead:

$$\mathcal{L}_{\text{Preference}}(\phi) := -\frac{1}{B} \sum_{i=1}^{B} \left[ \log \sigma(r_\phi(x_i, y_i^w) - r_\phi(x_i, y_i^l)) \right]. \tag{10}$$

**Debiasing Term Estimation.** Since the response pairs $(x, y, \bar{y})$ contains sufficient information to determine the bias attribute $b$, we can conclude that the RM forward process $(b \to (x, y, \bar{y}) \to H \to \mathbf{1}_{y \succ \bar{y}})$ is a Markov Chain (Shannon, 1948), where $H = [h_\phi(x, y), h_\phi(x, \bar{y})]$ is the last hidden states of the RM's transformer backbone. According to the data processing inequality (Shannon, 1948) and the CLUB upper bound (Cheng et al., 2020) in equation 6, we have

$$I(\mathbf{1}_{y \succ \bar{y}}; b) \leq I(H; b) \leq I_{\text{CLUB}}(H; b), \tag{11}$$

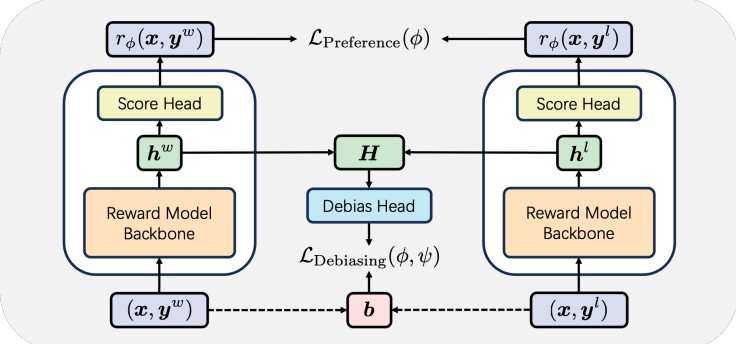

Figure 1: The proposed DIR framework. The architecture of the reward model is considered as a transformer backbone and an RM score head. The original RM ranking loss $\mathcal{L}_{\text{Preference}}(\phi)$ is calculated based on the outputs of the score head between each preference pair. The last hidden states $(\boldsymbol{h}^w, \boldsymbol{h}^l)$ of the backbone are collected as the representation $\boldsymbol{H}$. The debiasing loss $\mathcal{L}_{\text{debiasing}}(\phi, \psi)$ is computed between the inductive bias label $\boldsymbol{b}$ and the output of the debiasing head with parameter $\psi$.

where $I_{\text{CLUB}}(\boldsymbol{H}; \boldsymbol{b})$ can be practically calculated with a variational approximation network $q_\psi(\boldsymbol{b}|\boldsymbol{H})$ within the data batch $\mathcal{D}_{\text{Pref}} = \{(\boldsymbol{x}_i, \boldsymbol{y}_i^w, \boldsymbol{y}_i^l, \boldsymbol{b}_i) | \boldsymbol{y}_i^w \succ \boldsymbol{y}_i^l\}_{i=1}^B$:

$$-I_{\text{CLUB}}(\boldsymbol{H}; \boldsymbol{b}) \approx -\frac{1}{B} \sum_{i=1}^B \left[ \log q_\psi(\boldsymbol{b}_i|\boldsymbol{H}_i) - \frac{1}{B} \sum_{j=1}^B \log q_\psi(\boldsymbol{b}_j|\boldsymbol{H}_i) \right] =: \mathcal{L}_{\text{Debiasing}}(\phi, \psi), \quad (12)$$

where $\boldsymbol{H}_i = [\boldsymbol{h}_i^w, \boldsymbol{h}_i^l] = [\boldsymbol{h}_\phi(\boldsymbol{x}_i, \boldsymbol{y}_i^w), \boldsymbol{h}_\phi(\boldsymbol{x}_i, \boldsymbol{y}_i^l)]$. By minimizing $\mathcal{L}_{\text{Debiasing}}(\phi, \psi)$ as a upper bound estimation of $I(\boldsymbol{1}_{\boldsymbol{y} \succ \bar{\boldsymbol{y}}}; \boldsymbol{b})$, we effectively reduce the correlation between the biased attribute $\boldsymbol{b}$ and the RM hidden representation $\boldsymbol{h}_\phi(\boldsymbol{x}, \boldsymbol{y})$. Hence, the output RM scores $r_\phi(\boldsymbol{x}, \boldsymbol{y})$, as a deterministic function of $\boldsymbol{h}_\phi(\boldsymbol{x}, \boldsymbol{y})$, can remain unaffected from the inductive bias attributes $\boldsymbol{b}$.

As proved in Cheng et al. (2020), the better $q_\psi(\boldsymbol{b}_i|\boldsymbol{H}_i)$ approximates the ground-truth data distribution $p^*(\boldsymbol{b}_i|\boldsymbol{H}_i)$, the more accurate $I_{\text{CLUB}}$ serves as the MI upper bound estimator. Therefore, during the optimization of $\mathcal{L}_{\text{Debiasing}}(\phi, \psi)$ as in equation 12, we simultaneously maximize the log-likelihood of $q_\psi(\boldsymbol{b}_i|\boldsymbol{H}_i)$ within the batch samples $\{(\boldsymbol{H}_i, \boldsymbol{b}_i)\}_{i=1}^B$ to maintain the accuracy of the MI estimator:

$$\mathcal{L}_{\text{Estimator}}(\psi) := \frac{1}{B} \sum_{i=1}^B \log q_\psi(\boldsymbol{b}_i|\boldsymbol{H}_i). \quad (13)$$

**Overall Objective.** Based on the above discussion, the original RM debiasing objective in equation 8 converts to the following learning loss, which is practically tractable:

$$\min_\phi \mathcal{L}_{\text{Preference}}(\phi) + \lambda \cdot \mathcal{L}_{\text{Debiasing}}(\phi, \psi). \quad (14)$$

The illustration of the loss-calculation pipeline is shown in Figure 1. To make sure $\mathcal{L}_{\text{Debiasing}}(\phi, \psi)$ constantly have an accurate estimation to upper bound $I(\boldsymbol{1}_{\boldsymbol{y} \succ \bar{\boldsymbol{y}}}; \boldsymbol{b})$, we iteratively updated $r_\phi(\boldsymbol{x}, \boldsymbol{y})$ and $q_\psi(\boldsymbol{b}|\boldsymbol{H})$ within each training batch as shown in Algorithm 1. We refer to the proposed method as **D**ebiasing via **I**nformation optimization for **R**Ms (DIR).

## 4    RELATED WORK

**Reward Hacking of LLMs.** Reward hacking occurs when a policy model exploits spurious correlations or misspecifications within the reward function to achieve high scores without fulfilling the intended goal (Pan et al., 2022), which has emerged as a critical challenge in the stable and effective RL training of LLMs (Langosco et al., 2023; Hurst et al., 2024; Kaufmann et al., 2024; Skalse et al., 2025; Zhang et al., 2025; Li et al., 2025b). In RLHF, if the RM inadvertently learns inductive bias from the preference data (e.g., a bias towards more verbose (Singhal et al., 2023), or sycophantic responses (Sharma et al., 2023)), the LLM being optimized will learn to exploit these flaws, leading to a degradation in true performance (Hurst et al., 2024). Prior work has sought to mitigate reward hacking by empowering RMs, including better data curation (Liu et al., 2024a; Wang et al., 2025b; Dubois et al., 2024), model scaling up (Wang et al., 2025b), model ensembling (Wang et al., 2024), reward post-hoc calibration (Huang et al., 2024), causal inference (Shen et al., 2023; Wang et al., 2025a), and disentangled reward learning (Bu et al., 2025; Chen et al., 2024). Close to our method, InfoRM (Miao et al., 2024) employs an information-theoretic framework to compress the entire latent representation of the RM backbone, indirectly removing spurious information.

---

**Algorithm 1:** The iterative training processes of $r_\phi(\boldsymbol{x}, \boldsymbol{y})$ and $q_\psi(\boldsymbol{b}|\boldsymbol{H})$.

---

1  **Input**: Preference pairs with bias attributes $\mathcal{D}_{\text{Pref}} = \{(\boldsymbol{x}_i, \boldsymbol{y}_i^w, \boldsymbol{y}_i^l, \boldsymbol{b}_i)\}_{i=1}^N$, learning rates $\alpha_1, \alpha_2 > 0$;

2  **while** *each training iteration* **do**

3     Sample a batch of triplets $\{(\boldsymbol{x}_i, \boldsymbol{y}_i^w, \boldsymbol{y}_i^l, \boldsymbol{b}_i)\}_{i=1}^B \sim \mathcal{D}_{\text{Pref}}$;

4     Encode $(\boldsymbol{x}_i, \boldsymbol{y}_i^w)$ and $(\boldsymbol{x}_i, \boldsymbol{y}_i^l)$ into embeddings $\boldsymbol{H}_i = [\boldsymbol{h}_\phi(\boldsymbol{x}_i, \boldsymbol{y}_i^w), \boldsymbol{h}_\phi(\boldsymbol{x}_i, \boldsymbol{y}_i^l)]$;

5     **for** *bias estimator updating steps* **do**

6         Calculate the estimator loss $\mathcal{L}_{\text{Estimator}}(\psi) = -\frac{1}{B}\sum_{i=1}^B \log q_\psi(\boldsymbol{b}_i|\boldsymbol{H}_i)$;

7         Update approximation $q_\psi(\boldsymbol{b}|\boldsymbol{H})$ with $\psi \leftarrow \psi - \alpha_2 \cdot \nabla_\psi \mathcal{L}_{\text{Estimator}}(\psi)$;

8     **end**

9     Calculate RM preference loss $\mathcal{L}_{\text{Preference}}(\phi) = -\frac{1}{B}\sum_{i=1}^B [\log \sigma(r_\phi(\boldsymbol{x}_i, \boldsymbol{y}_i^w) - r_\phi(\boldsymbol{x}_i.\boldsymbol{y}_i^l))]$ ;

10    Calculate RM debiasing loss

     $\mathcal{L}_{\text{Debiasing}}(\phi, \psi) = -\frac{1}{B}\sum_{i=1}^B [\log q_\psi(\boldsymbol{b}_i|\boldsymbol{H}_i) - \frac{1}{B}\sum_{j=1}^B \log q_\psi(\boldsymbol{b}_j|\boldsymbol{H}_i)]$;

11    Compute RM total loss $\mathcal{L}_{\text{Total}}(\phi, \psi) = \mathcal{L}_{\text{Preference}}(\phi) + \lambda \cdot \mathcal{L}_{\text{Debiasing}}(\phi; \psi)$;

12    Update reward model $r_\phi$ with $\phi \leftarrow \phi - \alpha_1 \cdot \nabla_\phi \mathcal{L}_{\text{Total}}(\phi, \psi)$;

13  **end**

---

**Debiasing Methods of Language Models.** Debiasing methods seek to prevent models from learning and amplifying undesirable biases inherent in training data (He et al., 2019; Nam et al., 2020; Blodgett et al., 2020). The development of debiasing methods in natural language models has evolved from the word level (Caliskan et al., 2017; Kaneko & Bollegala, 2019; Manzini et al., 2019), to the sentence level (Liang et al., 2020; Cheng et al., 2021), and has gradually extended to generative LLMs (Wang et al., 2023; Gallegos et al., 2025), most of which focus on essential *social biases*, including gender (Kaneko & Bollegala, 2019; Fatemi et al., 2023), race (Caliskan et al., 2017), and age (Liu et al., 2024b). Core strategies for language model debiasing include adversarial training (Nam et al., 2020), causal inference (Zhou et al., 2023a), and information-theoretic methods (Tartaglione et al., 2021; Liu et al., 2023; Wang et al., 2023). Unlike generative LLM debiasing, the reward model debiasing methods focus on inductive bias attributes such as response length (Singhal et al., 2023), format (Zhang et al., 2025), and sycophancy (Denison et al., 2024). For instance, Chen et al. (2024); Bu et al. (2025) and Zhang et al. (2025) suppress length or format bias by penalizing the Pearson correlation between rewards and bias attributes, only capturing linear dependencies and missing higher-order interactions. Shen et al. (2023) use a two-head architecture for length bias but relies on heuristic disentanglement without explicitly modeling the preference–bias relationship. Wang et al. (2025a) enforce counterfactual invariance via MMD, which may over-constrain the reward model and distort its signal.

## 5  EXPERIMENT

We first evaluate the effectiveness of our DIR method under three practical debiasing scenarios: *response length*, *sycophancy*, and *format* as the inductive biases, respectively. Then, we explore whether our method can alleviate the concurrent multi-bias problem.

**Relative Bias Attributes.** In our DIR framework, to minimize the correlation between the biased attribute $\boldsymbol{b}$ and the RM hidden representation $\boldsymbol{H}$, the variational approximation of $q_\psi(\boldsymbol{b}|\boldsymbol{H})$ is required. However, when considering response length as the biased attribute, directly predicting the exact number of tokens in each response only based on the compressed representation $\boldsymbol{H}$ is very challenging. Therefore, instead of predicting the absolute value of response length, we introduce the *relative bias attributes*, which only consider the difference between chosen and rejected responses. For response length, the relative bias $\boldsymbol{b} = \mathbf{1}\{\text{length}(\boldsymbol{y}) > \text{length}(\bar{\boldsymbol{y}})\} \in \{0, 1\}$, indicating whether the chosen response is longer than the rejected one or not. Thus, the variational approximation for $q_\psi(\boldsymbol{b}|\boldsymbol{H})$ becomes a binary classifier indicating the label of the relative bias.

**Implementation Details.** Based on the above discussion, we have converted the response length bias into a relative binary bias indicator. Hence, under all three debiasing setups, the bias attributes can be represented by a categorical label, *e.g.*, "longer/shorter" for response length, and "sycophantic/in-sycophantic" for sycophancy. Therefore, in the experiments, we implement the variational network $q_\psi$ for bias estimation as a lightweight two-layer categorical classifier: $q_\psi(\boldsymbol{b}|\boldsymbol{H}) = \text{Softmax}(\text{MLP}(\boldsymbol{H}))$.

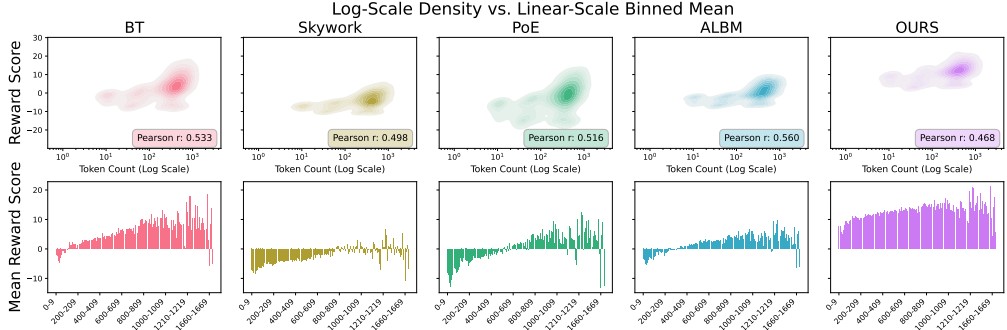

Figure 2: Evaluation of length bias in RMs on RM-Bench. We calculated the correlation between response length and reward score for RMs trained with different methods. Our approach yields the lowest Pearson correlation coefficient ($r = 0.468$), proving its effective ability in assigning more uniform reward scores.

Moreover, to respond to the relative bias, we further consider the linear transformation to the hidden $\boldsymbol{H} = [\boldsymbol{h}_\phi(\boldsymbol{x}, \boldsymbol{y}^w), \boldsymbol{h}_\phi(\boldsymbol{x}, \boldsymbol{y}^l)]$ as the *difference* $\Delta \boldsymbol{h} = \boldsymbol{h}_\phi(\boldsymbol{x}, \boldsymbol{y}^w) - \boldsymbol{h}_\phi(\boldsymbol{x}, \boldsymbol{y}^l)$ to emphasize the representation difference of the distinct features of the two responses.

**Training Setups.** Among all three RM debiasing scenarios, we use Llama3.1-8B-Instruct (Grattafiori & Team, 2024) as the reward model backbone and the initial checkpoint. Besides, we fully fine-tune the reward models with a global batch size of 128. The initial learning rate $\alpha_1$ for RMs is $2e-6$, which decays with a Cosine scheduler. For the bias estimator head, its learning rate $\alpha_2$ is set to $1e-3$. We fine-tune the policy models using Low-Rank Adaptation (LoRA) (Hu et al., 2021) with a global batch size of 128 and bfloat16 precision for one epoch. Both the actor and critic models use a learning rate of $1e-5$. The LoRA configuration employs a rank of 8 and an $\alpha$ of 32. The maximum context length is set to 4096, and the maximum generation length is set to 2048. The generation temperature for rollouts is set to 0.7.

## 5.1 LENGTH DEBIASING

**Datasets and Models.** We conduct the response length debiasing experiments by training RMs on the Skywork-Preference-80K-v0.2 (SK) dataset (Liu et al., 2024a). Then, we test the debiased RMs' performance in RLHF training with Llama3.1-8B-Instruct and OpenRLHF-Llama3-8B-SFT (Dong et al., 2024) as the initial policy and PPO (Schulman et al., 2017) as the learning algorithm for one epoch. PPO training prompts are 20K samples from the Alpaca-GPT4-EN dataset (Peng et al., 2023).

**Baselines and Evaluations.** We consider the following baselines for reproducibility: (1) Vanilla BT Baseline and popular open-source RM Skywork-Reward-Llama-3.1-8B-v0.2 (Liu et al., 2024a); (2) Length Debiased RMs, including PoE (Shen et al., 2023) and ALBM (Bu et al., 2025); (3) Length Penalty that directly resharps the reward during PPO by $\tilde{r}(\boldsymbol{x}, \boldsymbol{y}) = r(\boldsymbol{x}, \boldsymbol{y}) - 0.001 * \text{len}(\boldsymbol{y})$ (Dong et al., 2024); (4) InfoRM (Miao et al., 2024) that is also designed from the information theory perspective. Our evaluation protocol utlize few-shot settings for GSM8K (4-shot) (Cobbe et al., 2021), Race (3-shot) (Lai et al., 2017), and TriviaQA (5-shot) (Joshi et al., 2017). All other benchmarks, including Hellaswag (Zellers et al., 2019), IFeval (Zhou et al., 2023b), MMLU (Hendrycks et al., 2021), ProcessBench (Zheng et al., 2025), BBH (Suzgun et al., 2022), and HumanEval (Chen et al., 2021), are in a zero-shot setting. We report accuracy as the primary metric for all tasks.

**Reward Model Evaluation, Results and Analysis.** We first evaluate the inherent length bias of RMs by analyzing the correlation between their scores and response lengths on the RM-Bench (Liu et al., 2024c). As visualized in Figure 2, the standard BT RM exhibits a strong, undesirable positive correlation between length and reward (Pearson r = 0.533), which implies that even without an explicit preference for length in the training data[1], the model still learns a spurious "longer is better" heuristic, highlighting a fundamental issue in standard BT: the objective itself is susceptible to capturing such simple, non-causal patterns. Our approach demonstrates an effective ability to mitigate the length bias by achieving a Pearson correlation of just 0.468, the lowest among all evaluated methods. The quantitative advantage is further illustrated in the binned mean reward plots, where our flatter curve demonstrates the success of mitigating RM preferring longer responses. By learning to assign

---

[1]Average token number of $(\boldsymbol{x}, \boldsymbol{y}^w)$ in the SK training set is less than $(\boldsymbol{x}, \boldsymbol{y}^l)$ ones (622.86 vs. 707.24).

Table 1: RLHF performance based on different length-debiased RMs. **Bold** scores mean the best. Underline scores are the second-best. The $\Delta$ indicates the performance change relative to the respective Baseline.

| Benchmark | Llama3.1-8B-Instruct | | | | | | | OpenRLHF-Llama3-8B-SFT | | | | | | |
|---|---|---|---|---|---|---|---|---|---|---|---|---|---|---|
| | Base | SK | PoE | LP | ALBM | InfoRM | Ours | Base | SK | PoE | LP | ALBM | InfoRM | Ours |
| GSM8K | 83.93 | 84.61 | 83.62 | 75.97 | 84.08 | 83.78 | **84.84** | 74.83 | 78.17 | 77.79 | 77.18 | 78.85 | 76.74 | **79.08** |
| Hellaswag | 77.21 | 76.42 | 77.08 | 73.15 | 77.21 | 76.78 | **77.33** | 72.51 | **74.76** | 72.51 | 72.51 | 74.63 | 72.12 | 74.52 |
| IFeval | 72.83 | 70.06 | 71.72 | 65.47 | 73.57 | 74.12 | **78.00** | 44.92 | 45.10 | 49.72 | 46.21 | 46.21 | 46.21 | **52.31** |
| MMLU | 72.31 | 72.33 | 71.97 | 65.13 | 72.55 | 72.22 | **72.64** | 54.45 | 52.40 | 54.77 | 54.45 | **55.25** | 54.97 | 54.30 |
| ProcessBench | 25.39 | **29.49** | 28.50 | 24.91 | 26.12 | 26.25 | 27.73 | 4.46 | 10.31 | 9.68 | 7.84 | 10.85 | 3.24 | **13.82** |
| Race | 66.50 | 53.89 | 60.03 | **78.90** | 59.00 | 65.20 | 62.02 | 79.21 | 78.82 | **81.39** | 80.30 | 80.69 | 78.72 | 80.32 |
| BBH | 64.52 | 65.69 | 60.50 | 61.10 | 64.84 | 66.13 | **67.27** | 61.20 | 62.68 | 62.69 | 62.28 | 61.10 | 61.62 | **62.99** |
| HumanEval | **70.12** | 68.29 | 66.46 | 60.37 | 65.85 | **70.12** | **70.12** | 60.98 | 57.32 | 59.76 | 59.76 | 60.37 | 57.32 | **63.41** |
| TriviaQA | 32.64 | 49.01 | 48.41 | 47.20 | 52.09 | 30.56 | **55.86** | 48.53 | **52.86** | 52.34 | 48.32 | 51.52 | 48.16 | 52.52 |
| Avg. Acc. | 62.83 | 63.31 | 63.14 | 61.36 | 63.92 | 62.80 | **66.20** | 55.68 | 56.94 | 57.85 | 56.54 | 57.72 | 55.34 | **59.25** |
| $\Delta$ | - | ↑0.48 | ↑0.31 | ↓1.47 | ↑1.09 | ↓0.03 | ↑3.37 | - | ↑1.26 | ↑2.17 | ↑0.86 | ↑2.04 | ↓0.34 | ↑3.57 |

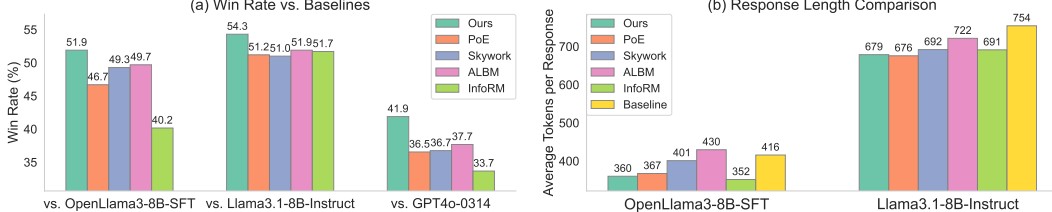

Figure 3: RLHF Evaluation on ArenaHard-v0.1 with different length-debiased RMs. (a) Head-to-head win rates. Policies are PPO fine-tuned from specified base models (from left to right: OpenLlama3-8B-SFT, Llama3.1-8B-Instruct, and Llama3.1-8B-Instruct, respectively) using five different RMs, which then act as challengers against opponents. (b) Average response length comparison.

scores more uniformly across different lengths, our method produces a more reliable RM, preventing the policy from being misguided into generating unnecessarily verbose outputs during subsequent fine-tuning. We report the performance on RM-Bench in Appendix C.1

**PPO Evaluation, Results and Analysis.** We report the RLHF performance based on the above length-debiased RMs across several benchmarks in Table 1, which demonstrates that mitigating length bias does not compromise, and ideally enhances, the policy model's core reasoning and knowledge capabilities. Using Llama3.1-8B-Instruct as the initial checkpoint, our method achieves the highest average performance of 66.20, significantly outperforming strong baselines. Furthermore, the trend of improved performance is consistent across different base models, as our method also secures the top average score on the OpenRLHF-Llama3-8B-SFT backbone, indicating that our fine-tuning strategy successfully enhances objective performance by mitigating the length bias. We also assess the user preference for policies fine-tuned using different reward models and compare average response length on the ArenaHard-v0.1 benchmark (Li et al., 2024). Figure 3 shows the head-to-head win rates of these challenger policies against strong opponents, as judged by Qwen3-235B-A22B-2507 (Yang et al., 2025). The policy trained with our method consistently demonstrates the highest win rate across all conditions. For instance, in Figure 3 (a), when fine-tuned on Llama3.1-8B-Instruct, ours achieves a remarkable 54.3% win rate against the baseline and 41.9% against GPT-4o-0314. Crucially, Figure 3 (b) reveals that the improved preference is achieved with expected conciseness. The policy guided by our RM produces shorter responses (e.g., 679 tokens on the Llama3.1 base) compared to policies guided by other RMs like ALBM (722 tokens) and the verbose original baseline (754 tokens). The better trade-off of a relatively higher win rate and lower verbosity shows that DIR can successfully guide PPO to produce a more efficient and human-aligned policy, effectively overcoming the common "longer is better" bias. In addition, we report the Win Rate performance on MT-Bench (Zheng et al., 2023) and Length-Control Alpaca (Dubois et al., 2025) in Appendix C.1, from which we observe that DIR can yield policies that are preferred more often.

**Additional Experimental Results.** We visualize the PPO training dynamics metrics, such as RLHF Rewards and KL divergences in Appendix C.1, which demonstrates that our RM helps make PPO training more stable with higher rewards. We analyze the training cost in Appendix C.2, which shows that the significant performance improvements do not come at the expense of prohibitive computational costs. We provide a detailed ablation study on $\lambda$ and representation difference in Appendix C.3, where the performance demonstrates the trade-off between preference learning and debiasing, showing better effectiveness of representation difference than concatenation. We also provide a case study in Appendix E.

Table 2: Evaluation on ArenaHard-v0.1 for policies fine-tuned with DPO, DPO+LC and DPO+Ours.

| OpenRLHF-Llama-3-8B-SFT | | |
|---|---|---|
| vs Base | Win Rate (%) | Length |
| DPO | 38.63 | 436.55 |
| +LC | 40.96 | 407.23 |
| +Ours | **45.27** | **404.61** |
| Meta-Llama3.1-8B-Instruct | | |
| vs Base | Win Rate (%) | Length |
| DPO | 44.06 | 700.87 |
| +LC | 46.57 | 691.43 |
| +Ours | **49.09** | **684.67** |

Table 3: Length debiasing performance via DPO training. **Bold** is best, underline is second-best.

| Benchmark | Meta-Llama3.1-8B-Instruct | | | OpenRLHF-Llama3-8B-SFT | | |
|---|---|---|---|---|---|---|
| | DPO | +LC | +Ours | DPO | +LC | +Ours |
| GSM8K-4shots | 82.11 | 81.43 | **82.56** | 75.89 | 76.35 | **77.26** |
| Hellaswag | 74.84 | **75.17** | 75.10 | 66.56 | 66.41 | **73.91** |
| IFeval | 73.57 | 74.12 | **74.68** | 38.26 | 35.86 | **41.59** |
| MMLU | 71.36 | **71.58** | 71.54 | 48.92 | 48.92 | **57.01** |
| ProcessBench | 26.75 | 26.70 | **27.60** | 4.28 | 4.95 | **6.93** |
| Race-3shots | 69.98 | 70.14 | **70.29** | 79.27 | 78.68 | **79.97** |
| BBH | **67.22** | 66.81 | 66.99 | 59.73 | 60.36 | **62.34** |
| HumanEval | 62.80 | 65.24 | **69.51** | 59.15 | **60.37** | 59.15 |
| TriviaQA-5shots | 55.12 | 55.15 | **55.29** | 47.45 | 47.69 | **48.93** |
| Avg. | 64.86 | 65.04 | **65.95** | 53.28 | 53.29 | **60.12** |
| Δ | - | ↑ 0.18 | ↑ 1.09 | - | ↑ 0.01 | ↑ 6.84 |

**Combination with Direct Preference Optimization (DPO).** In addition to PPO, Direct Preference Optimization (DPO) (Rafailov et al., 2024a) has emerged as a powerful post-alignment method that directly trains the policy to increase the log-probability of the preferred response relative to the rejected one. Here, we explore whether our DIR can be effectively combined with DPO. Conceptually, DIR operates at the reward modeling stage and should not modify the DPO objective: DPO still optimizes the standard log-sigmoid preference loss, and our method only modifies preference signals by making them less correlated with inductive bias. Specifically, we add $\lambda\mathcal{L}_{\text{Debiasing}}(\phi,\psi)$ to DPO loss. Empirically, we conduct the corresponding experiments, which show that our method can also improve DPO's performance with controlled length. We provide training details in Appendix C.4. Results in Table 2 indicate that our method leads to a final policy with both a better win-rate and more effective length control, effectively boosting vanilla DPO and also outperforming a specialized Length Controlled DPO (DPO+LC) variant (Park et al., 2024). Results in Table 3 indicate that our method also leads to a final policy with performance gains, especially for the SFT model that has not undergone a preference alignment. In summary, experiments on Table 2 and Table 3 suggest that debiased reward signals from DIR interact smoothly with DPO and effectively remove spurious gradients induced by length bias.

## 5.2 SYCOPHANCY DEBIASING

**Datasets, Models, and Baselines.** Sycophancy bias occurs when an RM learns to favor responses that agree with or flatter the user, rather than prioritizing factual accuracy and helpfulness. Motivated by Sharma et al. (2023); Wang et al. (2025a), we create a semi-sycophantic dataset by partially contaminating the HelpSteer3 dataset (Wang et al., 2025b). Specifically, we artificially inject a sycophantic prefix (i.e., "Yes, you are right.") into a proportion $\gamma$ (e.g., $\gamma = 40\%$) of responses in the training dataset. Within the contaminated subset, the prefix appears in the chosen response with probability $\alpha$ (e.g., $\alpha = 70\%$) and in the rejected response with probability $1 - \alpha = 30\%$, where the remaining $1 - \gamma = 60\%$ of the dataset remains unmodified and contains no sycophantic phrases. This contamination process creates a challenging, mixed-distribution environment in which the sycophantic phrase serves as a strong but unreliable reward signal. Since other debiasing methods are either mainly designed for length bias (e.g., PoE (Shen et al., 2023), ALBM (Bu et al., 2025), and Length-Penalty (Dong et al., 2024)) or are not open-sourced (e.g., CRM (Wang et al., 2025a)), we primarily compare our method against two key baselines: a standard BT reward model (Bradley & Terry, 1952) and InfoRM (Miao et al., 2024).

**Evaluation, Results, and Analysis.** To evaluate the models' susceptibility to sycophancy, we conduct an adversarial test. We take a clean evaluation set and create two versions: a "natural" version and a "sycophantic" version where the undesirable prefix is added to the rejected responses. We then measure the model's accuracy in correctly identifying the preferred response in both scenarios. A robust model should maintain its accuracy, whereas a biased model's performance will degrade when faced with the "flattering but wrong" responses. As shown in Table 4, the performance of the reward models varies under different settings. The BT model shows vulnerability to the bias, as its accuracy on natural examples is generally the lowest, particularly under high contamination. While InfoRM shows a clear improvement and greater resilience, our method demonstrates the most consistent and robust performance, which frequently achieves the highest accuracy across natural, adversarial, and overall settings, even under high contamination ratios. In summary, Table 4 indicates that our explicit debiasing mechanism is effective at mitigating the influence of sycophantic signals, enabling the model to focus more on the intrinsic quality of the response. In addition,

Table 4: Preference accuracy of sycophancy-debiased RMs under different contamination settings.

| Settings | | All. | | | Nat. | | | Adv. | | |
|---|---|---|---|---|---|---|---|---|---|---|
| $\gamma$ | $\alpha$ | BT | InfoRM | Ours | BT | InfoRM | Ours | BT | InfoRM | Ours |
| 20% | 30% | 86.6 | 89.4 | **90.2** | 85.5 | 88.9 | **89.8** | 91.0 | 91.2 | **93.6** |
| 20% | 50% | 85.6 | **89.8** | 88.7 | 85.7 | **90.3** | 88.2 | 84.9 | 87.9 | **90.9** |
| 20% | 70% | 84.8 | 86.1 | **87.1** | 85.2 | 86.0 | **87.5** | 83.1 | **86.6** | 85.1 |
| 40% | 30% | 87.4 | 89.0 | **90.9** | 86.0 | **88.1** | 87.4 | 88.9 | 90.3 | **93.9** |
| 40% | 50% | 86.1 | 87.9 | **88.7** | 87.0 | 87.7 | **89.8** | 84.8 | 88.3 | **89.1** |
| 40% | 70% | 83.6 | 86.6 | **88.0** | 84.4 | 86.3 | **87.4** | 82.6 | 87.2 | **88.6** |
| 80% | 30% | 89.0 | 90.4 | **91.3** | 82.3 | 89.5 | **88.0** | 90.7 | 91.9 | **92.2** |
| 80% | 50% | 85.5 | 87.2 | **88.1** | 86.3 | 86.3 | **86.2** | 85.3 | 87.5 | **90.3** |
| 80% | 70% | 81.2 | 84.5 | **86.2** | 86.4 | 86.4 | **87.2** | 79.7 | 84.0 | **86.2** |

we experimentally find that DIR can effectively handle multple concurrent biases as detailed in Appendix C.5.

## 5.3 FORMAT DEBIASING

**Datasets, Models, and Baselines** Zhang et al. (2025); Long et al. (2024) have shown that format biases, such as the use of lists, emojis, and boldface, are prevalent in human annotations and strong preference models. Hence, we process a format-biased dataset following the data generation protocol of Zhang et al. (2025). The base preference dataset consists of 71.6K response pairs, obtained by filtering `UltraFeedback` (Cui et al., 2024) to retain only pairs with a human score difference exceeding 1.0. To introduce format bias, we augment this clean dataset with two types of synthetically biased examples: (1) 0.7% of training pairs where a response wrapped in bold formatting is spuriously labeled as preferred over an identical unformatted version, and (2) 1.4% of pairs where a list-formatted response is similarly assigned a false preference label. The overall training set combines the clean and biased subsets. We compare against three baselines under the same experimental setup as (Zhang et al., 2025): (i) standard BT, (ii) BT trained after removing all format-biased examples from the training data (denoted BT†), and (iii) the Format Decoupling (FD) method (Zhang et al., 2025).

**Evaluation, Results, and Analysis.** As reported in Table 5, the standard BT model exhibits pronounced format bias, achieving win-rates of 89.0% and 92.5% for responses in Bold and List formats, respectively, providing strong evidence that vanilla BT conflates superficial formatting cues with response quality. The BT† variant, while partially mitigating this bias through data filtering, suffers a substantial drop in downstream performance on RewardBench, indicating that naive removal of format-biased samples compromises the model's ability to learn robust reward signals. In contrast, both FD and our method successfully neutralize format bias, driving win-rates close to the ideal 50% thresh-

Table 5: RM performance on both Bold and List format debiasing.

| Metric | BT | BT† | FD | Ours |
|---|---|---|---|---|
| *Win-Rate (%)* | | | | |
| Bold | 89.0 | 49.0 | **50.5** | 51.2 |
| List | 92.5 | 52.5 | 53.0 | **52.0** |
| *RewardBench (Filtered)* | | | | |
| Chat | **98.3** | 92.2 | 97.2 | 93.0 |
| Chat Hard | 71.4 | 64.4 | 72.8 | **80.1** |
| Safety | 83.1 | 75.5 | 82.9 | **89.6** |
| Reasoning | 85.1 | 81.4 | 89.7 | **92.2** |

old. Crucially, our approach outperforms FD on the more challenging subsets of RewardBench, demonstrating superior generalization in high-stakes domains, which underscores that our method achieves a more favorable trade-off between format debiasing and preference learning.

## 6 CONCLUSION

We introduce **D**ebiasing via **I**nformation optimization of **R**Ms (DIR), a novel information-theoretic method, to address the pervasive issue of inductive biases in reward modeling for RLHF. By maximizing the mutual information between RM scores and genuine human preference signals while minimizing the mutual information between RM predictions and biased attributes, DIR effectively disentangles genuine human preference signals from spurious correlations, *e.g.*, response length, sycophancy, and format. Equipped with variational bounds and MI estimation strategies, our method handles non-linear and complex bias structures with theoretical rigor and practical efficacy. Extensive experiments across diverse LLM and reward model benchmarks demonstrate that DIR not only mitigates a broad spectrum of biases but also enhances the generalization and alignment quality of reward models, leading to more robust downstream performance. We believe DIR offers a principled, scalable, and widely applicable solution for building more reliable and balanced alignment systems, paving the way toward more human-value-consistent artificial intelligence.

## 7 ACKNOWLEDGMENT

This work is supported by Longgang District Special Funds for Science and Technology Innovation under Grant LGKCSDPT2023002.

## ETHICS STATEMENT

This work aims to enhance the fairness and reliability of LLMs by mitigating format biases, preventing models from "gaming" evaluations based on style over substance. Our method encourages a more accurate assessment of a model's true capabilities. We acknowledge that our method only addresses the specific format biases targeted during training and does not mitigate broader societal or demographic biases. Furthermore, our ablation studies show that an overly aggressive debiasing coefficient ($\lambda$) can create a trade-off, potentially harming performance on simpler tasks. While we use public models and datasets, we recognize they may contain their own inherent biases. We believe our contribution is a positive step towards more robust and transparent AI alignment.

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

# A USAGE OF LLMS

In the preparation of this paper, we utilized Large Language Models (LLMs) solely for the purpose of grammatical polishing and text refinement of the manuscript content. Specifically, the LLMs were only used to optimize the clarity, fluency, and grammatical accuracy of the written text.

All content polished by LLMs underwent thorough manual review and verification by the authors. We carefully checked the polished text to ensure its consistency with the original research intent, accuracy of scientific facts, and compliance with academic integrity standards. We confirm that we take full responsibility for all contents of the paper under our names, including the parts that underwent LLM-assisted grammatical polishing.

# B BOUND PROOF

## B.1 PROOF OF THE BARBER-AGAKOV (BA) BOUND

We aim to prove that for any variational distribution $q_\theta(\mathbf{y}|\mathbf{x})$, the mutual information $I(\mathbf{x}; \mathbf{y})$ is lower-bounded by:

$$I(\mathbf{x}; \mathbf{y}) \geq \mathbb{E}_{p(\mathbf{x},\mathbf{y})}[\log q_\theta(\mathbf{y}|\mathbf{x})] + H[p] =: I_{\text{BA}}(\mathbf{x}; \mathbf{y}), \tag{15}$$

where $H[p]$ is the entropy of the ground-truth distribution $p(\boldsymbol{x}, \boldsymbol{y})$. By the definition of mutual information, we have:

$$I(\mathbf{x}; \mathbf{y}) = H(\mathbf{y}) - H(\mathbf{y}|\mathbf{x}) = H(\mathbf{y}) + \mathbb{E}_{p(\mathbf{x},\mathbf{y})}[\log p(\mathbf{y}|\mathbf{x})]. \tag{16}$$

Consider the expected Kullback-Leibler (KL) divergence between the true conditional distribution $p(\mathbf{y}|\mathbf{x})$ and the variational approximation $q_\theta(\mathbf{y}|\mathbf{x})$, we have:

$$\mathbb{E}_{p(\mathbf{x})}\left[D_{\text{KL}}\big(p(\mathbf{y}|\mathbf{x}) \,\|\, q_\theta(\mathbf{y}|\mathbf{x})\big)\right] = \mathbb{E}_{p(\mathbf{x},\mathbf{y})}\left[\log \frac{p(\mathbf{y}|\mathbf{x})}{q_\theta(\mathbf{y}|\mathbf{x})}\right] \geq 0. \tag{17}$$

By the linearity of expectation, the inequality 17 can be rearranged as:

$$\mathbb{E}_{p(\mathbf{x},\mathbf{y})}[\log p(\mathbf{y}|\mathbf{x})] \geq \mathbb{E}_{p(\mathbf{x},\mathbf{y})}[\log q_\theta(\mathbf{y}|\mathbf{x})]. \tag{18}$$

Then we can substitute equation 18 into the definition of $I(\mathbf{x}; \mathbf{y})$, yielding:

$$I(\mathbf{x}; \mathbf{y}) \geq H[p] + \mathbb{E}_{p(\mathbf{x},\mathbf{y})}[\log q_\theta(\mathbf{y}|\mathbf{x})], \tag{19}$$

which completes the proof. The bound is tight if and only if $q_\theta(\mathbf{y}|\mathbf{x}) = p(\mathbf{y}|\mathbf{x})$ almost everywhere with respect to $p(\mathbf{x}, \mathbf{y})$.

## B.2 PROOF OF THE CLUB UPPER BOUND

We aim to prove that for any variational distribution $q_\theta(\boldsymbol{y}|\boldsymbol{x})$, the mutual information $I(\boldsymbol{x}; \boldsymbol{y})$ is upper-bounded by $I_{\text{CLUB}}(\boldsymbol{x}; \boldsymbol{y})$. We begin with the definition of mutual information:

$$I(\boldsymbol{x}; \boldsymbol{y}) = \mathbb{E}_{p(\boldsymbol{x},\boldsymbol{y})}\left[\log p(\boldsymbol{y}|\boldsymbol{x})\right] - \mathbb{E}_{p(\boldsymbol{y})}\left[\log p(\boldsymbol{y})\right] \tag{20}$$

Let's focus on the second term, which is the negative marginal entropy $+H(\boldsymbol{y})$. We can express the marginal distribution $p(\boldsymbol{y})$ by marginalizing out $\boldsymbol{x}$:

$$p(\boldsymbol{y}) = \mathbb{E}_{p(\boldsymbol{x}')}[p(\boldsymbol{y}|\boldsymbol{x}')] \tag{21}$$

where $\boldsymbol{x}'$ is a random variable drawn from the same distribution as $\boldsymbol{x}$, but is independent of the $\boldsymbol{x}$ in the first term of equation 20. Substituting equation 21 into the entropy term:

$$-\mathbb{E}_{p(\boldsymbol{y})}\left[\log p(\boldsymbol{y})\right] = -\mathbb{E}_{p(\boldsymbol{y})}\left[\log \mathbb{E}_{p(\boldsymbol{x}')}[p(\boldsymbol{y}|\boldsymbol{x}')]\right]. \tag{22}$$

Since the logarithm is a concave function, we can apply Jensen's inequality, which states that $\mathbb{E}[\log(Z)] \leq \log(\mathbb{E}[Z])$ and implies $-\log(\mathbb{E}[Z]) \leq -\mathbb{E}[\log(Z)]$. Applying this, we get:

$$-\mathbb{E}_{p(\boldsymbol{y})}\left[\log \mathbb{E}_{p(\boldsymbol{x}')}[p(\boldsymbol{y}|\boldsymbol{x}')]\right] \leq -\mathbb{E}_{p(\boldsymbol{y})}\left[\mathbb{E}_{p(\boldsymbol{x}')}[\log p(\boldsymbol{y}|\boldsymbol{x}')]\right] = -\mathbb{E}_{p(\boldsymbol{x}')p(\boldsymbol{y})}[\log p(\boldsymbol{y}|\boldsymbol{x}')]. \tag{23}$$

Now, substituting equation 23 back into our original MI expression equation 20, we obtain an upper bound on the mutual information:

$$I\left(\boldsymbol{x};\boldsymbol{y}\right) \leq \mathbb{E}_{p(\boldsymbol{x},\boldsymbol{y})}[\log p(\boldsymbol{y}|\boldsymbol{x})] - \mathbb{E}_{p(\boldsymbol{x})p(\boldsymbol{y})}[\log p(\boldsymbol{y}|\boldsymbol{x})]. \tag{24}$$

Note that the second expectation is over the product of marginals $p(\boldsymbol{x})p(\boldsymbol{y})$. The inequality equation 24 holds for the true conditional distribution $p(\boldsymbol{y}|\boldsymbol{x})$. The CLUB bound replaces $p(\boldsymbol{y}|\boldsymbol{x})$ with the variational approximation $q_\theta(\boldsymbol{y}|\boldsymbol{x})$. The key insight from Cheng et al. (2020) is that the difference between the true bound and the variational bound is an expectation of KL-divergences, and the proposed variational form serves as a practical, sample-based upper bound for minimization. Therefore, we use the variational form as our tractable objective:

$$I\left(\boldsymbol{x};\boldsymbol{y}\right) \leq \mathbb{E}_{p(\boldsymbol{x},\boldsymbol{y})}[\log q_\theta(\boldsymbol{y}|\boldsymbol{x})] - \mathbb{E}_{p(\boldsymbol{x})p(\boldsymbol{y})}[\log q_\theta(\boldsymbol{y}|\boldsymbol{x})] =: I_{\text{CLUB}}(\boldsymbol{x};\boldsymbol{y}), \tag{25}$$

which completes the justification for using $I_{\text{CLUB}}$ as an upper bound for mutual information minimization.

## C    Experiment

### C.1    Length Bias

**Training Settings.**    For our PPO experiment, we fine-tune two distinct models using 20,000 samples from the alpaca-gpt4-data-en dataset (Peng et al., 2023). The first model, Llama3.1-8B-Instruct (Grattafiori & Team, 2024), has undergone post-training that includes both DPO and RLHF. The second, OpenRLHF-Llama3-8B-SFT (Dong et al., 2024), is an instruction-following version built upon Llama3-8B-Base, without the RLHF post-training stage. We conduct the PPO training using the ms-swift (Zhao et al., 2025) framework with its default PPO training configuration.

**Baselines.**    We mainly consider the following baselines due to the reproducibility: (1) Vanilla BT Baseline and popular open-source RM Skywork-Reward-Llama-3.1-8B-v0.2 (Liu et al., 2024a); (2) Length Debiased RMs, including PoE (Shen et al., 2023) and ALBM (Bu et al., 2025); (3) Length Penalty that directly resharps the reward during PPO by $\tilde{r}(\boldsymbol{x},\boldsymbol{y}) = r(\boldsymbol{x},\boldsymbol{y}) - 0.001 * len(\boldsymbol{y})$ (Dong et al., 2024); (4) InfoRM (Miao et al., 2024) that is also designed from the information theory perspective.

**Performance on RM-Bench.**    We further evaluate our debiased reward models on RM-Bench (Liu et al., 2024c), a comprehensive benchmark that assesses model capabilities across four distinct domains, including `Chat`, `Math`, `Code`, and `Safety` with three difficulty levels: `Hard`, `Normal`, and `Easy`. As shown in Table 6, our DIR framework consistently outperforms several strong baselines in terms of overall performance. Our primary variant, denoted as Ours-1.0, corresponds to the optimal trade-off point identified in our ablation study ($\lambda = 1.0$), which achieves the second-highest aggregate score of 69.35, reflecting a well-calibrated balance between debiasing and generalization and indicating that DIR enhances the reward model's discriminative capacity on core reasoning tasks without substantially degrading its general-purpose alignment.

When we increase the debiasing strength to $\lambda = 10.0$, the resulting model Ours-10.0 achieves the highest overall score of 70.18. The most pronounced improvement occurs on the Hard subset, where performance surges to 64.41, surpassing the next-best method by over 16 points. Such a substantial performance improvement suggests that by explicitly suppressing reliance on superficial bias through the DIR mechanism, the reward model is better able to attend to nuanced, content-based indicators of response quality, particularly those that are critical for evaluating complex or challenging prompts. Moreover, Ours-10.0 achieves the top scores in both the `Chat` and `Code` domains. However, this stronger debiasing comes at a cost: performance on the `Easy` subset declines relative to weaker debiasing settings. On such instances, where simple heuristics often suffice for accurate judgment, the aggressive removal of bias signals appears overly restrictive and counterproductive. In summary, these results demonstrate that DIR not only enhances the overall capability of the reward model but also offers a tunable mechanism to prioritize robustness on challenging tasks over simpler ones, howcasing the flexibility and effectiveness of our approach.

Table 6: Performance comparison on RM-Bench. Best results are in **bold**. Second-performance is underlined.

| Method | Chat | Math | Code | Safety | Hard | Normal | Easy | Total |
|---|---|---|---|---|---|---|---|---|
| BT | 64.69 | 61.21 | 51.41 | 95.11 | 42.76 | 72.30 | 89.24 | 68.10 |
| PoE | 67.70 | 61.23 | 51.51 | **95.51** | 44.94 | 73.17 | 88.86 | 68.99 |
| ALBM | 64.57 | 58.48 | 52.34 | 95.21 | 47.88 | 71.50 | **90.32** | 67.40 |
| Ours-1.0 | 68.91 | **61.81** | 51.56 | 95.13 | 47.88 | **73.59** | 88.93 | 69.35 |
| Ours-10.0 | **71.23** | 61.59 | **52.73** | 94.91 | **64.41** | 71.29 | 74.85 | **70.18** |

Table 7: Win rate (%) performance comparison on MT-Bench.

| Win Rate (%) (vs. Base) | Base Model | |
|---|---|---|
| | OpenRLHF-Llama-3-8B-SFT | Meta-Llama3.1-8B-Instruct |
| Ours | **56.25** | **56.88** |
| PoE | 48.75 | 51.25 |
| Skywork | 49.38 | 51.25 |
| ALBM | 53.75 | 50.63 |
| InfoRM | 46.88 | 51.88 |

**Performance on MT-Bench and AlpaceEval.** For MT-Bench (Zheng et al., 2023), we report the win rate of each RM-guided policy against its own base model, using the standard MT-Bench LLM-as-a-judge setup. As shown in Table 7, our method ("Ours") achieves the highest win rates on both backbones (56.25% vs. 48.75–53.75% for OpenRLHF-Llama-3-8B-SFT, and 56.88% vs. 50.63–51.88% for Meta-Llama3.1-8B-Instruct), indicating more improvements on open-ended, multi-turn dialogue quality.

For Length Controlled AlpacaEval, we follow the length-controlled protocol of Dubois et al. (2025) and report both raw win rate and length-controlled win rate over the base model. On Meta-Llama3.1-8B-Instruct, Ours achieves the highest scores on both metrics. On OpenRLHF-Llama-3-8B-SFT, Skywork attains a slightly higher raw win rate, but Ours achieves the best length-controlled win rate, which is consistent with our goal: once the confounding effect of response length is controlled for, our debiased RMs yield policies that are preferred more often, demonstrating better alignment that is not driven by verbosity. We will include these MT-Bench and AlpacaEval results and their analysis in the revised version.

**PPO Training Monitoring.** Figure 4 presents three key metrics for monitoring the PPO training process. The left plot (RLHF Reward) evaluates the final quality score of the model's outputs, with higher values being better. The middle plot (KL Divergence) measures how much the learned policy has deviated from the initial reference model, indicating the extent of exploration. The right plot (Approx. KL) shows the magnitude of each policy update, serving as a critical indicator of training stability. Our policy model demonstrates a better balance across these metrics by achieving a top reward score that significantly outperforms all baselines. Concurrently, our KL divergence is maintained at a moderate level, suggesting effective exploration without catastrophic deviation from the base model's capabilities. Most importantly, our method exhibits the lowest and most stable Approx. KL, which proves that the training process is exceptionally smooth and reliable. In summary, our approach successfully boosts performance while ensuring unparalleled training stability.

## C.2 RM TRAINING COST ANALYSIS.

We analyze the computational overhead in terms of GPU memory consumption and training time, with a detailed comparison presented in Table 9. We use 8 GPU cards with full parameter training and DeepSpeed Zero-1(Rajbhandari et al., 2020). Our approach demonstrates highly comparable resource efficiency to existing methods. Specifically, the GPU memory usage of our method (57.22GB) is only marginally higher than the baseline (56.80GB) and on par with other techniques like ALBM (56.88GB). Regarding training time, while our method (67.09 minutes) requires a moderate increase compared to the simpler baseline (50.46 minutes), DIR remains competitive and aligns closely with other advanced methods such as ALBM (68.21 minutes), which shows that the significant

Table 8: Performance comparison on Length Controlled AlpacaEval against `gpt4-1106-preview`.

| Methods | Raw Win Rate (%) | Length Control Win Rate (%) |
|---|---|---|
| **Base Model: Meta Llama3.1-8B-Instruct** | | |
| Ours | **31.30** | **19.66** |
| PoE | 26.58 | 11.41 |
| Skywork | 29.38 | 13.21 |
| ALBM | 26.83 | 10.61 |
| InfoRM | 25.22 | 11.02 |
| **Base Model: OpenRLHF Llama-3-8B-SFT** | | |
| Ours | 9.50 | **5.46** |
| PoE | 7.14 | 3.28 |
| Skywork | **10.19** | 3.93 |
| ALBM | 8.88 | 5.08 |
| InfoRM | 5.84 | 3.65 |

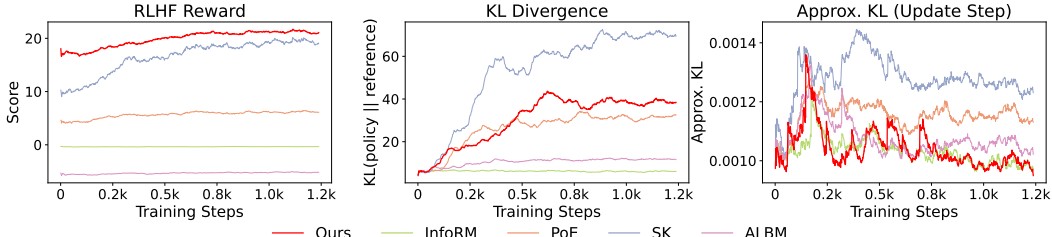

Figure 4: PPO training dynamics across key metrics. Our RM obtains a higher policy score and demonstrates better training stability.

performance improvements offered by our approach do not come at the expense of prohibitive computational costs, establishing our approach as a practical and efficient solution.

## C.3 Ablation Studies under Length Debias

**Ablation Study on Representation for Debiasing.** In this section, we investigate the influence of difference-based representation $\Delta h = h^w - h^l$ as input to the variational debiasing network $q_\psi$, compared with the representation of the concatenating form $[h^w; h^l]$. We compare both variants on `RewardBench-v1` and `RM-Bench`. Results in Table 10 show that the difference-based variant slightly outperforms the concatenation-based one across most domains and difficulty levels. Empirically, the difference-based approach yields clear gains. On `RewardBench-v1`, accuracy on `Chat Hard` improves from 78.9% to 83.6%, and on `Reasoning` from 88.8% to 90.0%. On `RM-Bench`, the `Chat` score increases from 63.9% to 66.8%. Performance on other subsets remains stable, indicating no trade-off in generalization. From an efficiency standpoint, the difference operator preserves the embedding dimensionality, whereas concatenation doubles it. The reduced input size lowers both parameter count and GPU memory usage during training. Given its empirical advantage, theoretical grounding, and computational efficiency, we adopt representation difference as the default input formulation in DIR.

**Ablation Study on Debiasing Coefficient $\lambda$.** The hyperparameter $\lambda$ in Equation 14 governs the trade-off between the standard preference learning objective ($\mathcal{L}_{\text{Preference}}$) and our information-theoretic debiasing objective ($\mathcal{L}_{\text{Debiasing}}$). To analyze its sensitivity, we tested a range of values: $\lambda \in \{0.1, 0.3, 0.5, 1, 2, 5, 10\}$. The results, visualized in Figure 5, reveal a clear trade-off.

As shown in the figure, when $\lambda$ is too small (e.g., 0.1), the debiasing signal is insufficient. The model behaves similarly to a standard BT model, exhibiting a high bias metric (e.g., high Pearson correlation with a bias attribute) while achieving good performance on RewardBench. Conversely, when $\lambda$ is

Table 9: Training cost comparison.

| Method | GPU Memory | Training Time |
|---|---|---|
| Baseline | 55.08GB | 50.46m |
| PoE | 56.80GB | 55.35m |
| ALBM | 57.22GB | 78.21m |
| InfoRM | 57.99GB | 75.21m |
| Ours | 56.88GB | 67.09m |

Table 10: Ablation study on the representation format for the debiasing module. We report accuracy (%) on RewardBench-v1 and RM-Bench. The difference-based approach consistently outperforms concatenation, especially on challenging conversational and reasoning tasks. Best results are in **bold**.

| | RewardBench-v1 (Acc %) | | | | RM-Bench (Acc %) | | | |
|---|---|---|---|---|---|---|---|---|
| Method | Chat | Chat Hard | Safety | Reasoning | Chat | Math | Code | Safety |
| Concat ($[\boldsymbol{h}^w; \boldsymbol{h}^l]$) | 93.3 | 78.9 | **90.9** | 88.8 | 65.9 | 60.8 | **52.6** | 95.0 |
| Difference ($\Delta \boldsymbol{h}$) | **94.1** | **83.6** | 89.7 | **90.0** | **67.8** | **61.1** | 52.4 | **95.2** |

too large (e.g., 10), the debiasing objective dominates the training. The resulting "over-correction" successfully minimizes the bias metric but severely compromises the reward model's ability to learn true preference signals, leading to a significant drop in RewardBench accuracy. Moreover, we observe that $\lambda = 1$ strikes an optimal balance. At $\lambda = 1$, the bias metric is substantially reduced, while the preference learning performance on RewardBench is maximized. The behavior demonstrates that the proposed method can effectively neutralize spurious correlations without damaging—and in fact while enhancing—the reward model's core capabilities. Therefore, all main experiments in this paper use $\lambda = 1$.

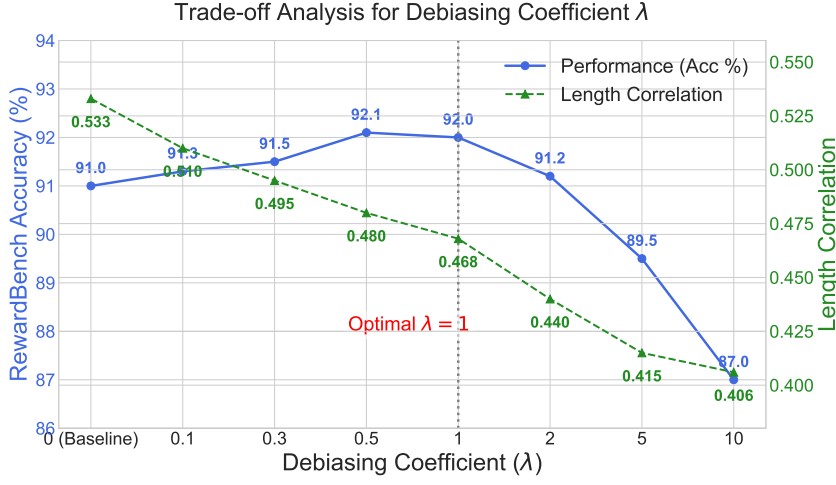

Figure 5: Ablation study on the debiasing coefficient $\lambda$. The plot shows the trade-off between preference learning performance (RewardBench Accuracy, blue) and the bias metric (e.g., Pearson $r$, green). $\lambda = 1$ achieves the best balance.

## C.4 EXPERIMENT ON DPO

Specifically, we adopt the ms-swift framework with its default DPO training configuration on Human-Like-DPO-Dataset (Çalık & Akkuş, 2025), based on both OpenRLHF-Llama-3-8b-SFT and Meta-Llama3.1-8B-Instruct models, where DPO $\beta = 0.1$, debias factor $\lambda = 1$. We train 1 epoch and evaluate the performance on the final checkpoint. Human-Like-DPO-Dataset is created to fine-tune LLMs toward generating more human-like responses, which includes 10,884 samples across 256 topics, covering technology, daily Life, science, history and arts. We evaluate the performance

on ArenaHard-v0.1 and several popular benchmarks. For baselines, we also compare with the length-controlled DPO method (Park et al., 2024), which disentangles the length from the quality to explicitly avoid the policy model from preferring the longer response DPO training.

## C.5 CONCURRENT MULTI-BIASES

Real-world datasets frequently exhibit multiple concurrent biases. Hence, we investigate whether DIR can effectively mitigate such co-occurring biases, specifically, length bias and sycophancy bias, simultaneously. Concretely, we extend DIR to length and sycophancy biases by introducing two independent mutual information terms, each with its own debiasing head $\psi_{\text{length}}$ and $\psi_{\text{syco}}$, respectively:

$$\mathcal{L}_{\text{Total}}(\phi) = \mathcal{L}_{\text{Preference}}(\phi) + \lambda_{\text{length}} \cdot \mathcal{L}_{\text{Debiasing}}(\phi, \psi_{\text{length}}) + \lambda_{\text{syco}} \cdot \mathcal{L}_{\text{Debiasing}}(\phi, \psi_{\text{syco}}), \quad (26)$$

where $\lambda_{\text{len}}$ and $\lambda_{\text{syco}}$ are set to 1.0. We follow the setting in Table 4, training Llama-3.1-8B-Instruct on the HelpSteer3 dataset under two sycophancy contamination configurations ($\gamma = 40\%$ and $80\%$, $\alpha = 70\%$), where a larger $\gamma$ indicates a more challenging setting. All models (BT, length-only DIR, and length+syco DIR) are trained for 1 epoch on the same data, and we report results using the final checkpoint for fairness and convenience.

As shown in Table 11, on RM-Bench, the joint model (Ours-Len-Syco) achieves the best overall performance and the largest gains on the hardest subset (e.g., at $\gamma = 40\%$, Total: $67.25 \rightarrow 69.96$, Hard: $39.88 \rightarrow 46.85$ vs. BT), while still clearly reducing the Pearson correlation with length relative to BT, confirming that length bias is mitigated even when sycophancy is also debiased. We also observe that the length-only model (Ours-Len) attains the lowest length–reward Pearson coefficient, but somewhat surprisingly, the joint length+sycophancy debiasing (Ours-Len-Syco) yields the best overall RM-Bench performance, suggesting that debiasing multiple biases together could also help lead to a more balanced reward model. As shown in Table 12, on sycophancy stress tests, debiasing only sycophancy (Ours-Syco) gives the strongest syco robustness, as expected, but the joint model (Ours-Len-Syco) still substantially outperforms BT on all sycophancy metrics (All./Nat./Adv.) across both $\gamma$ settings, while additionally reducing length bias. In summary, a second debiasing term leads to a controlled trade-off, not conflicting gradients: both biases are improved over BT, and overall RM quality remains strong.

Table 11: Performance comparison of concurrent multi-bias experiments on RM-Bench. Best results are in **bold**.

| $\gamma = 40\%, \alpha = 70\%$ | Chat | Math | Code | Safety | Hard | Normal | Easy | Total | Pearson Coefficient |
|---|---|---|---|---|---|---|---|---|---|
| BT | 66.58 | 64.17 | 53.70 | 84.53 | 39.88 | 72.82 | **89.04** | 67.25 | 0.4807 |
| Ours-Len | 66.93 | 64.59 | 53.12 | **89.14** | 44.25 | 72.43 | 88.65 | 68.44 | **0.4235** |
| Ours-Len-Syco | **70.80** | **65.07** | **55.26** | 88.71 | **46.85** | **75.08** | 87.96 | **69.96** | 0.4446 |

| $\gamma = 80\%, \alpha = 70\%$ | Chat | Math | Code | Safety | Hard | Normal | Easy | Total | Pearson Coefficient |
|---|---|---|---|---|---|---|---|---|---|
| BT | 65.37 | 63.71 | 53.12 | 80.85 | 37.58 | 70.96 | **88.75** | 65.76 | 0.4666 |
| Ours-Len | **68.91** | 64.25 | 53.75 | 87.23 | 44.78 | **73.09** | 87.72 | 68.53 | **0.4081** |
| Ours-Len-Syco | 68.39 | **64.92** | **54.04** | **88.06** | **45.15** | 72.92 | 88.50 | **68.85** | 0.4235 |

Table 12: Reward model accuracy (%) on the concurrent multi-bias under different contamination settings.

| $\gamma$ | $\alpha$ | All. BT | All. Ours-Syco | All. Ours-Len-Syco | Nat. BT | Nat. Ours-Syco | Nat. Ours-Len-Syco | Adv. BT | Adv. Ours-Syco | Adv. Ours-Len-Syco |
|---|---|---|---|---|---|---|---|---|---|---|
| 40% | 70% | 83.6 | 88.0 | 85.6 | 84.4 | 87.4 | 86.4 | 82.6 | 88.6 | 84.4 |
| 80% | 70% | 81.2 | 86.2 | 85.9 | 86.4 | 87.2 | 86.6 | 79.7 | 86.2 | 85.1 |

## D PROMPT-BASED JUSTIFICATION PROMPT

We provide a Qwen3-235B-A22B-based pair-wise justification prompt shown below, which is adopted from ArenaHard's official implementation.

*Please act as an impartial judge and evaluate the quality of the responses provided by two AI assistants to the user prompt displayed below. You will be given assistant A's answer and assistant B's answer. Your job is to evaluate which assistant's answer is better. Begin your evaluation by generating your own answer to the prompt. You must provide your answers before judging any answers. When evaluating the assistants' answers, compare both assistants' answers with your answer. You must identify and correct any mistakes or inaccurate information. Then consider if the assistant's answers are helpful, relevant, and concise. Helpful means the answer correctly responds to the prompt or follows the instructions. Note that when a user prompt has any ambiguity or more than one interpretation, it is more helpful and appropriate to ask for clarifications or more information from the user than providing an answer based on assumptions. Relevant means all parts of the response closely connect or are appropriate to what is being asked. Concise means the response is clear and not verbose or excessive. Then consider the creativity and novelty of the assistant's answers when needed. Finally, identify any missing important information in the assistants' answers that would be beneficial to include when responding to the user prompt. After providing your explanation, you must output only one of the following choices as your final verdict with a label:*

*1. Assistant A is significantly better: [[A >> B]]*
*2. Assistant A is slightly better: [[A>B]]*
*3. Tie, relatively the same: [[A=B]]*
*4. Assistant B is slightly better: [[B>A]]*
*5. Assistant B is significantly better: [[B>>A]]*
*Example output: "My final verdict is tie: [[A=B]]"*

*[User Prompt]*

*[Assistant A's Answer]*
*{answer A}*

*[Assistant B's Answer]*
*{answer B}*

# E  CASE STUDY

Our model demonstrates its ability to generate not just a correct, but a more professionally helpful response compared to strong baselines like GPT-4o. As visualized in Figure 6 (Data UID: `95bb5011680c4ba281ad7830ffa57f92`), when asked how to set up passwordless SSH, our model provided a guide that was praised by the LLM-based judge for its clear structure and organization. In this instance, GPT-4o's response is judged to be better to all other baselines except for our response, which provides a more secure, modern, and concise solution, highlighting its advanced capabilities.

Crucially, the judge noted that our model successfully avoids unnecessary verbosity, resulting in a response that is of higher quality and more concise. This is not merely a stylistic advantage; the substance of our model's response was also technically useful, which recommended the more modern and secure "ed25519" encryption algorithm over the RSA suggested by GPT-4o. Furthermore, our model included the critical security best practice of disabling password authentication entirely via the sshd_config file—a vital step for server hardening that GPT-4o and other models omitted.

**Prompt:** How do I set up ssh into a server such that I don't require a password when I ssh into it?

**ALBM (2486 chars):** Setting up Passwordless SSH**\n\nTo set up passwordless SSH, you'll need to generate a pair of SSH keys on your local machine and add the public key to the server. This way, you can authenticate without entering a password.\n\n### Step 1: Generate SSH Keys on Your Local Machine\n\nOpen a terminal on your local machine and run the following command:\n\n```bash\nssh-keygen -t rsa -b 4096\n```\n\nThis will generate a pair of SSH keys in the `~/.ssh` directory.\n\n### Step 2: Copy the Public Key to the Server\n\nYou'll need to copy the public key to the server. You can do this by running the following command:\n\n```bash\nssh-copy-id user@server\n```\n\nReplace `user` with your username on the server and `server` with the hostname or IP address of the server.\n\n### Step 3: Verify Passwordless SSH\n\nTry connecting to the server using SSH without a password:\n\n```bash\nssh user@server\n```\n\nIf everything is set up correctly, you should be able to connect to the server without entering a password.\n\n### Step 4: Add the Public Key to the Server (Alternative Method)\n\nIf you're using an older version of SSH or don't have `ssh-copy-id` available, you can add the public key manually. Run the following commands on your local machine:\n\n```bash\nssh-keygen -t rsa -b 4096\ncat ~/.ssh/id_rsa.pub | ssh user@server \"mkdir -p ~/.ssh; cat >> ~/.ssh/authorized_keys\"\n```\n\nThis will create the `~/.ssh` directory on the server if it doesn't exist and append the public key to the `authorized_keys` file.\n\n### Tips and Precautions\n\n* Make sure to keep your private key secure. If someone gets access to your private key, they can connect to your server without a password.\n* Use a strong passphrase for your private key to prevent unauthorized access.\n* If you're using a shared server or a server with multiple users, consider using a more secure method, such as public key authentication with a keyring or a secure authentication service.\n* Regularly update your SSH keys to ensure you're using the latest version.\n\n### Example Use Case\n\nHere's an example of how you can use passwordless SSH to automate a deployment script:\n\n```bash\n#!/bin/bash\n\n# Set up SSH connection\nssh -o \"StrictHostKeyChecking=no\" user@server <<EOF\n# Run deployment script\ngit pull origin main\nnpm install\nnpm run build\nEOF\n```\n\nIn this example, the script uses passwordless SSH to connect to the server and run a deployment script without prompting for a password.

**InfoRM (2422 chars):** To set up SSH to not require a password, you'll need to generate a pair of SSH keys (public and private) and add the public key to the server. Here's a step-by-step guide:\n\n### Step 1: Generate SSH keys\n\n1. Open a terminal on your local machine.\n\n2. Run the following command to generate a new pair of SSH keys:\n\n```bash\nssh-keygen -t ed25519\n```\n\n or\n\n```bash\nssh-keygen -t rsa\n```\n\n You can choose the type of key (RSA or Ed25519) and the location where the keys will be stored. By default, it will be stored in `~/.ssh`.\n\n### Step 2: Add the public key to the server\n\n1. Copy the public key to the clipboard:\n\n```bash\nncat ~/.ssh/id_ed25519.pub\n```\n\n or\n\n```bash\nncat ~/.ssh/id_rsa.pub\n```\n\n Copy the entire output.\n\n2. Log in to your server using your current password.\n\n3. Create a new directory for the SSH keys if it doesn't exist:\n\n```bash\nmkdir -p ~/.ssh\n```\n\n4. Add the public key to the authorized_keys file:\n\n```bash\necho \"your-public-key-here\" >> ~/.ssh/authorized_keys\n```\n\n Replace `your-public-key-here` with the public key you copied earlier.\n\n### Step 3: Restrict SSH access (optional)\n\nTo add an extra layer of security, you can restrict SSH access to specific users or IP addresses. You can do this by editing the `~/.ssh/authorized_keys` file and adding options to the key.\n\nFor example, to restrict access to a specific user and IP address, you can add the following options:\n\n```bash\nssh-rsa your-public-key-here user@ip_address\n```\n\nThis will only allow the specified user to access the server from the specified IP address.\n\n### Step 4: Update SSH configuration (optional)\n\nTo use the SSH keys for passwordless login, you'll need to update the SSH configuration. Edit the SSH configuration file:\n\n```bash\nsudo nano /etc/ssh/sshd_config\n```\n\n2. Add the following line to the end of the file:\n\n```bash\nPubkeyAuthentication yes\n```\n\n3. Restart the SSH service:\n\n```bash\nsudo service ssh restart\n```\n\n or\n\n```bash\nsudo systemctl restart sshd\n```\n\n### Step 5: Test SSH connection\n\nTry to SSH into the server using the following command:\n\n```bash\nssh user@server\n```\n\nIf everything is set up correctly, you should be able to log in without entering a password.\n\nNote: Make sure to keep your private key secure and do not share it with anyone. If your private key is compromised, you'll need to generate a new pair of SSH keys.

**PoE (2172 chars):** **Setting up SSH Key-Based Authentication**\n\nTo set up SSH key-based authentication, you'll need to generate a pair of SSH keys on your local machine and then add the public key to the server's authorized_keys file. This way, you can access the server without entering a password.\n\n**Step 1: Generate SSH Keys on Your Local Machine**\n\nOpen a terminal and run the following command to generate a new pair of SSH keys:\n\n```bash\nssh-keygen -t rsa -b 4096\n```\n\nThis will create a new pair of keys in the `~/.ssh` directory.\n\n* `id_rsa`: the private key (do not share this with anyone)\n* `id_rsa.pub`: the public key (share this with the server administrator)\n\n**Step 2: Copy the Public Key to the Server**\n\nCopy the contents of the public key file (`id_rsa.pub`) to the server's authorized_keys file using the following command:\n\n```bash\nssh-copy-id user@server\n```\n\nReplace `user` with your username on the server and `server` with the hostname or IP address of the server.\n\n**Alternative Method: Manually Copying the Public Key**\n\nIf you prefer not to use `ssh-copy-id`, you can manually copy the public key to the server's authorized_keys file using the following steps:\n\n1. Open the public key file (`id_rsa.pub`) in a text editor:\n\n```bash\nncat ~/.ssh/id_rsa.pub\n```\n\n2. Connect to the server using SSH:\n\n```bash\nssh user@server\n```\n\n3. Append the public key to the authorized_keys file:\n\n```bash\necho \"your_public_key_here\" >> ~/.ssh/authorized_keys\n```\n\nReplace `your_public_key_here` with the contents of the public key file.\n\n**Step 3: Test SSH Connection**\n\nTry connecting to the server using SSH without a password:\n\n```bash\nssh user@server\n```\n\nYou should now be able to access the server without entering a password.\n\n**Security Considerations**\n\n* Keep your private key secure and do not share it with anyone.\n* Limit access to the `authorized_keys` file and the `~/.ssh` directory to prevent unauthorized access.\n* Regularly update your SSH keys and authorized_keys file to ensure security.\n\nBy following these steps, you can set up SSH key-based authentication and access your server without entering a password.

**Skywork (2274 chars):** **Setting up SSH without Password**\n\nTo set up SSH without requiring a password, you'll need to generate public-private key pairs and configure SSH on both your local machine and the server.\n\n### Step 1: Generate Public-Private Key Pair on Local Machine\n\n1. **Open a terminal** on your local machine.\n\n2. **Generate a new key pair** using the following command:\n\n```bash\nssh-keygen -t ed25519\n```\n\n or\n\n```bash\nssh-keygen -t rsa\n```\n\n (Choose a key type: `ed25519` or `rsa`)\n\n3. **Accept the default filename and location** for the key pair.\n\n### Step 2: Copy the Public Key to the Server\n\n1. **Copy the public key** to the clipboard:\n\n```bash\nncat ~/.ssh/id_ed25519.pub\n```\n\n or\n\n```bash\nncat ~/.ssh/id_rsa.pub\n```\n\n2. **Log in to the server** using your existing SSH password.\n\n3. **Create a new directory** for SSH keys if it doesn't exist:\n\n```bash\nmkdir -p ~/.ssh\n```\n\n4. **Append the public key** to the server's authorized_keys file:\n\n```bash\necho \"<paste the public key here>\" >> ~/.ssh/authorized_keys\n```\n\n5. **Set permissions** to ensure only the owner can read the authorized_keys file:\n\n```bash\nchmod 700 ~/.ssh\nchmod 600 ~/.ssh/authorized_keys\n```\n\n### Step 3: Configure SSH on the Server\n\n1. **Edit the SSH configuration file** to enable public key authentication:\n\n```bash\nsudo nano /etc/ssh/sshd_config\n```\n\n2. **Add or uncomment the following line**:\n\n```\nPubkeyAuthentication yes\n```\n\n3. **Restart the SSH service** to apply the changes:\n\n```bash\nsudo systemctl restart sshd\n```\n\n### Step 4: SSH into the Server without a Password\n\n1. **Try to SSH into the server** using the private key:\n\n```bash\nssh -i ~/.ssh/id_ed25519 user@server_ip\n```\n\n or\n\n```bash\nssh -i ~/.ssh/id_rsa user@server_ip\n```\n\nYou should now be able to SSH into the server without entering a password!\n\n**Additional Tips:**\n\n* Make sure to keep your private key secure and do not share it with anyone.\n* Consider using a passphrase-protected private key for added security.\n* Regularly update your SSH keys to ensure you're using the latest encryption algorithms.\n\nBy following these steps, you'll be able to set up SSH without a password and enjoy a more secure and convenient way to access your servers.

**OURS (2052 chars):** **Generating SSH Keys**\n\nTo set up SSH without a password, you'll need to generate a pair of SSH keys: a private key and a public key. The private key will be kept on your local machine, while the public key will be added to the server.\n\n**Step 1: Generate SSH Keys**\n\nOpen a terminal on your local machine and run the following command:\n\n```bash\nssh-keygen -t ed25519\n```\n\nThis will generate a new pair of SSH keys using the Ed25519 algorithm. You can choose a different algorithm (e.g., RSA) if you prefer.\n\n**Step 2: Copy the Public Key**\n\nOnce the keys are generated, you'll see a message indicating that the keys have been created. The public key will be stored in a file named `id_ed25519.pub` (or `id_rsa.pub` if you chose RSA).\n\nTo copy the public key to the server, run the following command:\n\n```bash\nssh-copy-id user@server\n```\n\nReplace `user` with your username on the server and `server` with the hostname or IP address of the server.\n\n**Step 3: Authorize the Public Key**\n\nOn the server, the public key will be added to the `~/.ssh/authorized_keys` file. You can verify this by running:\n\n```bash\ncat ~/.ssh/authorized_keys\n```\n\nYou should see the public key from your local machine listed.\n\n**Step 4: Test SSH without a Password**\n\nNow, try connecting to the server using SSH without a password:\n\n```bash\nssh user@server\n```\n\nIf everything is set up correctly, you should be logged in without being prompted for a password.\n\n**Tips and Considerations**\n\n* Make sure to keep your private key secure, as it can be used to access your account on the server.\n* Consider using a passphrase for your private key to add an extra layer of security.\n* If you're using a shared hosting environment, check with your provider to see if they have any specific requirements or restrictions for SSH key usage.\n* To disable password authentication on the server, you can edit the `sshd_config` file and set `PasswordAuthentication` to `no`. Then, restart the SSH service.\n\nBy following these steps, you should be able to set up SSH without a password and enjoy a more convenient and secure way to access your server.

**GPT4o (2008 chars):** To set up SSH key-based authentication and avoid entering a password each time you SSH into a server, follow these steps:\n\n1. Generate an SSH key pair on your local machine (if you haven't already):\n\n Open a terminal and run the following command:\n\n```\n ssh-keygen -t rsa -b 4096 -C \"your_email@example.com\"\n```\n\n Replace \"your_email@example.com\" with your actual email address. This will create a 4096-bit RSA key pair.\n\n You'll be prompted to enter a file path to save the key pair. Press Enter to use the default location (usually `~/.ssh/id_rsa`).\n\n You'll also be prompted to enter a passphrase for the key pair. You can either enter a passphrase for added security or leave it empty for passwordless authentication.\n\n2. Copy the public key to the remote server:\n\n Use the `ssh-copy-id` command to copy your public key to the remote server:\n\n```\n ssh-copy-id username@remote_server_ip\n```\n\n Replace \"username\" with your remote server's username and \"remote_server_ip\" with the remote server's IP address or hostname.\n\n If `ssh-copy-id` is not available, you can manually copy the public key by using the following command:\n\n```\n cat ~/.ssh/id_rsa.pub | ssh username@remote_server_ip \"mkdir -p ~/.ssh && chmod 700 ~/.ssh && cat >> ~/.ssh/authorized_keys && chmod 600 ~/.ssh/authorized_keys\"\n```\n\n This command will create the `~/.ssh` directory and `authorized_keys` file on the remote server if they don't exist, and append your public key to the `authorized_keys` file.\n\n3. Test the passwordless SSH connection:\n\n Now, try to SSH into the remote server:\n\n```\n ssh username@remote_server_ip\n```\n\n If everything is set up correctly, you should be able to log in without entering a password. If you set a passphrase for your key pair, you'll be prompted to enter it.\n\nNote: Make sure to keep your private key (`~/.ssh/id_rsa`) secure and do not share it with anyone. The public key (`~/.ssh/id_rsa.pub`) is the one you copy to the remote server.

**Judgment:** My final verdict is [[A (OURS) >>B (GPT4o) ]]\n\n**Explanation:**\nBoth assistants correctly outline SSH key-based authentication setup. However, Assistant A's response is superior due to:\n1. **Modern Algorithm Recommendation**: Ed25519 is more secure and efficient than RSA (Assistant B's choice).\n2. **Security Best Practices**: Explicitly mentions disabling password authentication via `sshd_config`, a critical security step Assistant B omits.\n3. **Conciseness & Structure**: Clear step-by-step formatting with helpful tips (e.g., passphrases, provider checks) without unnecessary verbosity.\n4. **Permissions Handling**: Includes explicit `chmod` commands in examples, ensuring proper directory/file permissions.\n\nAssistant B's manual key-copy method is useful but lacks A's comprehensive security guidance. While both answers are technically correct, Assistant A provides a more secure, future-proof solution with better operational safety recommendations.

Figure 6: A case study on ArenaHard-v0.1 Benchmark where our model's response significantly outperforms GPT-4o on a technical question about setting up passwordless SSH.

