# OpenReview forum: "Eliminating Inductive Bias in Reward Models with Information-Theoretic Guidance"
_ICLR.cc/2026/Conference — ICLR 2026 Poster_

### Official Review · Reviewer_UoY4 · 2025-10-23

**Soundness:** 3
**Presentation:** 3
**Contribution:** 3
**Rating:** 6
**Confidence:** 4

**Summary:**

This paper targets to the issue of inductive bias of reward models. This paper proposes an information-theoretic debiasing method, called DIR, which maximizes the mutual information between preference prediction and input pairs, while minimizing the mutual information between outputs and predefined bias attributes to debiase these biases. Experimental results demonstrate its effectiveness in debiasing biases of length, sycophancy and format, while achieving competitive alignment performance.

**Strengths:**

- The paper is well written and clear to read.
- Denoising reward model is very important for RLHF to improve alignment performance of LLMs.
- The experimental results are solid, covering three types of biases.

**Weaknesses:**

- In the experimental setup, the authors address the biases of length, sycophancy, and format individually. However, in practice, we would expect a reward model to debias all these biases simultaneously. It seems that the paper lacks an experimental setup that assesses and discusses the ability of DIR to mitigate all these biases simultaneously.
- There are two commonly used benchmarks for preference alignment, i.e., AlpacaEval [1] and MT-Bench [2], which should be included to evaluate the model ability of open-ended and multi-turn generations.
- I think that a figure to illustrate your methods will help readers intuitively understand your proposed method.
- The proposed method DIR relies on predefined inductive biases, such as length, sycophancy and format. However, if the reward model has some potential biases that are not intuitive such as length perceived by human, can the proposed DIR handle such scenario?

[1] Length-controlled alpacaeval: A simple way to debias automatic evaluators.
[2] Judging llm-as-a-judge with mt-bench and chatbot arena.

**Questions:**

- Do you compare performance with DPO directly trained using the same preference data?
- What is your implementation details of the variational network $\psi$?

---

> ### Author Response · Authors · 2025-11-20
> **Response to Weakness-1: Concurrent Multi-Biases**
>
> We thank the reviewer for this insightful question and fully agree that real datasets often exhibit multiple concurrent biases. We conducted a new multi-bias experiment where DIR simultaneously debiases both length and sycophantic responses. Overall, we find that **the multi-bias DIR reduces both length and sycophancy bias compared to the BT baseline, indicating that multiple debiasing terms can be combined without significant destructive optimization conflicts.**
>
> Concretely, we extend DIR to these two explicit biases by introducing two independent Mutual-Information regularizers, each with its own $q_\psi$ head. For each preference pair, we construct the corresponding two bias labels, and optimize:
>
> $$
> \mathcal{L} = \mathcal{L}_{\text{reward}} + \lambda_{\text{len}} * \mathcal{L}_{\text{debias_len}} + \lambda_{\text{syco}} * \mathcal{L}_{\text{debias_syco}}
> $$
>
> where $\lambda_{\text{len}} = \lambda_{\text{syco}}=1$. We follow the Table 3 setting, training Meta-Llama-3.1-8B-Instruct on HelpSteer3 dataset under two sycophancy contamination configurations ($\gamma = 0.4 / 0.8, \alpha = 0.7$), where a larger $\gamma$ indicates a more challenging setting. All models (BT, length-only DIR, and length+syco DIR) are trained for 1 epoch on the same data, and we report results using the final checkpoint for fairness and convenience.
>
> As shown in Table 1, on RM-Bench, the joint model (OURS-Len-Syco) achieves the best overall performance and the largest gains on the hardest subset (e.g., at $\gamma=0.4$, Total: $67.25 \rightarrow 69.96$, Hard: $39.88 \rightarrow 46.85$ vs. BT), *while still clearly reducing the Pearson correlation with length relative to BT, confirming that length bias is mitigated even when sycophancy is also debiased*. We also observe that the length-only model (OURS-Len) attains the lowest length–reward Pearson coefficient, but somewhat surprisingly, the joint length+sycophancy debiasing (OURS-Len-Syco) yields the best overall RM-Bench performance, suggesting that *debiasing multiple biases together may help lead to a more balanced and effective reward model.*
>
> Table 1. RM-Bench performance on the length bias
>
> |  $\gamma=40$%, $\alpha=70$%     | Chat  | Math  | Code  | Safety | Hard  | Normal | Easy  | Total | Pearson Coefficient |
> |-------------------|-------|-------|-------|--------|-------|--------|-------|-------|---------------------|
> | BT                | 66.58 | 64.17 | 53.70 | 84.53  | 39.88 | 72.82  | **89.04** | 67.25 | 0.4807              |
> | OURS-Len          | 66.93 | 64.59 | 53.12 | **89.14**  | 44.25 | 72.43  | 88.65 | 68.44 | **0.4235**              |
> | OURS-Len-Syco     | **70.80** | **65.07** | **55.26** | 88.71  | **46.85** | **75.08**  | 87.96 | **69.96** | 0.4446              |
>
> |  $\gamma=80$%, $\alpha=70$%      | Chat  | Math  | Code  | Safety | Hard  | Normal | Easy  | Total | Pearson Coefficient |
> |-------------------|-------|-------|-------|--------|-------|--------|-------|-------|---------------------|
> | BT                | 65.37 | 63.71 | 53.12 | 80.85  | 37.58 | 70.96  | **88.75** | 65.76 | 0.4666              |
> | OURS-Len          | **68.91** | 64.25 | 53.75 | 87.23  | 44.78 | **73.09**  | 87.72 | 68.53 | **0.4081**              |
> | OURS-Len-Syco     | 68.39 | **64.92** | **54.04** | **88.06**  | **45.15** | 72.92  | 88.50 | **68.85** | 0.4235              |
>
>
> As shown in Table 2, on sycophancy stress tests, debiasing only sycophancy (OURS-Syco) gives the strongest syco robustness, as expected, but the joint model (OURS-Syco-Len) still *substantially outperforms BT on all sycophancy metrics (All/Nat/Adv) across both $\gamma$ settings, while additionally reducing length bias.*
>
> Table 2. Reward model accuracy (%) on the sycophancy bias
>
> | $\gamma$   | $\alpha$    | All. BT | All. OURS-Syco | All. OURS-Syco-Len | Nat. BT | Nat. OURS-Syco | Nat. OURS-Syco-Len | Adv. BT | Adv. OURS-Syco | Adv. OURS-Syco-Len |
> |------|------|---------|----------------|---------------------|--------|----------------|---------------------|---------|----------------|---------------------|
> | 40%  | 70%  | 83.6    | 88.0           | 85.6                | 84.4   | 87.4           | 86.4                | 82.6    | 88.6           | 84.4                |
> | 80%  | 70%  | 81.2    | 86.2           | 85.9                | 86.4   | 87.2           | 86.6                | 79.7    | 86.2           | 85.1                |
>
>
> In summary, a second debiasing term leads to a controlled trade-off, not conflicting gradients: both biases are improved over BT, and overall RM quality remains strong. We will complete the experiments and analysis, add these results in our revised paper.

---

> ### Author Response · Authors · 2025-11-20
> **Response to Weakness-2: Performance on AlpacaEval and MT-Bench; Weakness-3: Figure**
>
> **Response to Weakness-2: Performance on AlpacaEval and MT-Bench**
>
> We thank the reviewer for this helpful suggestion. Following your recommendation, we have additionally evaluated the PPO-finetuned policies on both MT-Bench and AlpacaEval.
>
> For MT-Bench, we report the win rate of each RM-guided policy against its own base model, using the standard LLM-as-a-judge setup. For convenience, we used the checkpoints as the same as those used on ArenaHard. As shown below, our method (“OURS”) achieves the highest win rates on both backbones (56.25% vs. 48.75–53.75% for OpenRLHF-Llama-3-8B-SFT, and 56.88% vs. 50.63–51.88% for Meta-Llama3.1-8B-Instruct), **indicating stronger improvements on open-ended, multi-turn dialogue quality.**
>
> Table 1. MT-Bench Win Rate, vs. Base Model.
>
> | Win Rate (%) MT-Bench (vs Base) | OpenRLHF-Llama-3-8B-SFT | Meta-Llama3.1-8B-Instruct |
> |---------------------------------|--------------------------|---------------------------|
> | OURS                            | **56.25**                    | **56.88**                     |
> | PoE                             | 48.75                    | 51.25                     |
> | Skywork                         | 49.38                    | 51.25                     |
> | ALBM                   | 53.75                    | 50.63                     |
> | InfoRM                          | 46.88                    | 51.88                     |
>
> For Length Controlled AlpacaEval, we follow the length-controlled protocol of [1] and report both raw win rate and length-controlled win rate over the base model. On Meta-Llama3.1-8B-Instruct, OURS achieves the highest scores on both metrics. On OpenRLHF-Llama-3-8B-SFT, Skywork attains a slightly higher raw win rate, but OURS achieves the best length-controlled win rate. **This pattern is consistent with our goal: once the confounding effect of response length is controlled for, our debiased RMs yield policies that are preferred more often, demonstrating better alignment that is not driven by verbosity.**
>
> Table 2. Length-Control AlpaceEval Win Rate, vs. gpt4_1106_preview.
>
> | Comparison                               | Raw Win Rate (%) | Length Control Win Rate (%) |
> |------------------------------------------|-------------------|-----------------------------|
> | Base Model: Meta Llama3.1-8B-Instruct    |                   |                             |
> | OURS                                     | **31.30**             | **19.66**                       |
> | PoE                                      | 26.58             | 11.41                       |
> | Skywork                                  | 29.38             | 13.21                       |
> | ALBM                             | 26.83             | 10.61                       |
> | InfoRM                                   | 25.22             | 11.02                       |
> | Base Model: OpenRLHF Llama-3-8B-SFT      |                   |                             |
> | OURS                                     | 9.50              | **5.46**                        |
> | PoE                                      | 7.14              | 3.28                        |
> | Skywork                                  | **10.19**             | 3.93                        |
> | ALBM                             | 8.88              | 5.08                        |
> | InfoRM                                   | 5.84              | 3.65                        |
>
>  We will include these MT-Bench and AlpacaEval results and their analysis in the revised version.
>
> [1] Length-controlled alpacaeval: A simple way to debias automatic evaluators.
>
> **Response to Weakness-3: Figure**
>
> Thanks for your constructive suggestion to help polish our manuscript! We will add a figure to illustrate our framework more clearly in our revised paper.

---

> ### Author Response · Authors · 2025-11-20
> **Response to Weakness-4: Predefined Bias**
>
> We thank the reviewer for highlighting this crucial point. We respectfully argue that requiring a pre-defined bias attribute $\boldsymbol{b}$ is a design choice, rather than an inherent limitation of DIR. Our goal is to achieve precise, verifiable debiasing on important alignment issues while remaining practical for real-world RLHF.
>
> 1. Precision and reliability vs. unsupervised approaches. **Conceptually**, by explicitly specifying a bias $\boldsymbol{b}$ and minimizing $I(\Delta \boldsymbol{h}; \boldsymbol{b})$, DIR gives a clear and auditable handle on which spurious signal is suppressed and how strongly (via $\lambda$). In contrast, unsupervised methods like InfoRM compress the representation without supervision on any specific bias and may still preserve informative but spurious cues if they correlate with true preference, or conversely, compress away information that is actually important for capturing human quality judgments. In such cases, even if the bias is partially reduced, the resulting reward model may not be effective for RLHF, which is also reflected in our results **empirically**: Table 1 shows that InfoRM does not consistently improve over the base models on either backbone, while DIR yields the largest average gains. In Figure 2(b), InfoRM reduces response length to a similar level as ours, but Figure 2(a) shows noticeably lower win rates, indicating *a weaker trade-off between debiasing and performance*. By directly targeting the known bias, DIR is mainly designed to remove the spurious component while preserving preference-relevant information, thereby achieving both better direct debiasing and higher preference quality.
>
> 2. High practicality in RLHF. In many RLHF applications, various significant problematic reward-hacking behaviors (length, formatting artifacts, sycophantic phrases, preference style) are usually known and can be perceived, thus trivial to label or compute. For these cases, specifying $\boldsymbol{b}$ or $\boldsymbol{b}_\text{rel}$ is cheap, and DIR provides a targeted, efficient way to neutralize exactly those biases we care about.
>
> 3. Alignment with prior work. Our setting follows the mainstream paradigm in both reward-hacking work [1–5] and border NLP debiasing [6–9], where defining the attribute to be neutralized (e.g., length, format, gender, race) is a prerequisite for effective intervention. DIR extends this paradigm with an information-theoretic framework for reward models: once $\boldsymbol{b}$ is defined, we offer a principled way to reduce its mutual information (MI) with the preference signal.
>
> In summary, we position DIR as a practical tool for some known, high-impact biases, complementary to more exploratory unsupervised methods. We will clarify this setting in our revised related work.
>
> [1] From lists to emojis: How format bias affects model alignment
>
> [2] Beyond excess and deficiency: Adaptive length bias mitigation in reward models for rlhf.
>
> [3]  Odin: Disentangled reward mitigates hacking in rlhf
>
> [4] Beyond reward hacking: Causal rewards for large language model alignment
>
> [5] Inform: Mitigating reward hacking in rlhf via information-theoretic reward modeling
>
> [6] Examining gender and race bias in two hundred sentiment analysis systems.
>
> [7] FAIRFIL: CONTRASTIVE NEURAL DEBIASING METHOD FOR PRETRAINED TEXT ENCODERS
>
> [8] Measuring and mitigating unintended bias in text classification.
>
> [9] On Measuring Social Biases in Sentence Encoders

---

> ### Author Response · Authors · 2025-11-20
> **Response to Question-1: DPO**
>
> In our main PPO experiments, the preference data used to train the reward models is different from that for PPO, so a strict comparison with DPO trained on exactly the same preference pairs may not be directly available. However, to address the reviewer’s concern, we additionally ran experiments where we combined DIR with DPO. These results show that **DPO+OURS not only alleviates the well-known “DPO prefers longer responses” issue, but also yields consistent improvements over vanilla DPO and a specialized length-controlled DPO variant.**
>
> Specifically, we adopt ms-swift [1] with its default DPO training configuration on Human-Like-DPO-Dataset [2], based on both OpenRLHF-Llama-3-8b-SFT and Meta-Llama3.1-8B-Instruct models, where DPO $\beta=0.1$, debias factor $\lambda=1$. We train 1 epoch and evaluate the performance on the final checkpoint. Human-Like-DPO-Dataset is created to fine-tune LLMs toward generating more human-like responses, which includes 10,884 samples across 256 topics, containing technology, daily Life, science, history and arts. We evaluate the performance on ArenaHard-v0.1 and popular benchmarks. For baselines, we also compare with length-controlled DPO method [3], which is designed to explicitly avoid the policy model from preferring the longer response during training. As shown in Table 1, we find that **our method can lead to a final policy that is both higher quality and more token-efficient:**
>
> Table 1. ArenaHard Win Rate (\%)
>
> | Win Rate (%)        | OpenRLHF-Llama-3-8B-SFT |                  | Meta-Llama3.1-8B-Instruct |                  |
> |---------------------|-------------------------|------------------|---------------------------|------------------|
> | vs Base             | ArenaHard-v0.1          | Length           | ArenaHard-v0.1            | Length           |
> | DPO                 | 38.63                   | 436.55           | 44.06                     | 700.87           |
> | DPO + LengthControl | 40.96                   | 407.23           | 46.57                     | 691.43           |
> | DPO + OURS          | **45.27**                   | **404.61**           | **49.09**                     | **684.67**           |
>
> Results on ArenaHard indicate that our method leads to a final policy with both a better win-rate and more effective length control, effectively boosting vanilla DPO and also outperforming a specialized Length Controlled DPO (DPO-LC) variant [3].
>
> Results on Table 2 indicate that **our method also leads to a final policy with performance gains, especially for SFT model, which has not undergone a preference alignment. **
>
> Table 1. Benchmark Performance
>
> | Benchmark                         | Meta-Llama3.1-8B-Instruct DPO | DPO-LC | DPO+OURS | OpenRLHF-Llama3-8B-SFT DPO | DPO-LC | DPO+OURS |
> |-----------------------------------|-------------------------------|--------|----------|-----------------------------|--------|----------|
> | GSM8K_acc-4shots                  | 82.11                         | 81.43  | 82.56    | 75.89                       | 76.35  | 77.26    |
> | Hellaswag_acc                     | 74.84                         | 75.17  | 75.10    | 66.56                       | 66.41  | 73.91    |
> | IFeval_acc                        | 73.57                         | 74.12  | 74.68    | 38.26                       | 35.86  | 41.59    |
> | MMLU_acc                          | 71.36                         | 71.58  | 71.54    | 48.92                       | 48.92  | 57.01    |
> | ProcessBench_acc                  | 26.75                         | 26.70  | 27.60    | 4.28                        | 4.95   | 6.93     |
> | Race_acc-3shots                   | 69.98                         | 70.14  | 70.29    | 79.27                       | 78.68  | 79.97    |
> | BBH_acc                           | 67.22                         | 66.81  | 66.99    | 59.73                       | 60.36  | 62.34    |
> | Humaneval_pass@1                  | 62.80                         | 65.24  | 69.51    | 59.15                       | 60.37  | 59.15    |
> | TriviaQA_acc-5shots               | 55.12                         | 55.15  | 55.29    | 47.45                       | 47.69  | 48.93    |
> | Avg. Performance                  | 64.86                         | 65.04  | 65.95    | 53.28                       | 53.29  | 60.12    |
> | Δ                                 | -                             | 0.18   | **1.09**     | -                           | 0.01   | **6.84**     |
>
> In summary, experiments in Table 1 and Table 2 suggest that **debiased reward signals from DIR interact smoothly with DPO and effectively remove spurious gradients induced by length bias**. We will include a more detailed analysis of these experiments in our revised paper and update our uploaded codebase to ensure full reproducibility.
>
> [1] https://github.com/modelscope/ms-swift
>
> [2] https://huggingface.co/datasets/HumanLLMs/Human-Like-DPO-Dataset
>
> [3] Disentangling Length from Quality in Direct Preference Optimization

---

> ### Author Response · Authors · 2025-11-20
> **Response to Question-2: Details of $q_\psi$**
>
> Our variational network $q_\psi$ for estimating the mutual information is implemented as a lightweight two-layer Multi-Layer Perceptron (MLP). Its architecture is as follows:
> ```python
> self.variational_net = nn.Sequential(
>     nn.Linear(input_dim, hidden_size),
>     nn.ReLU(),
>     nn.Linear(hidden_size, label_num)
> )
> ```
> The dimensions are chosen based on the following principles:
>
> input\_dim: This is dynamically set to match the dimension of the final hidden state representation from the backbone LLM. For instance, in our experiments with Llama-3.1-8B-Instruct, the input\_dim is 4096.
>
> label\_num: The output dimension is set to 2. This is a direct and necessary consequence of our theoretical formulation, as the relative bias attribute $\boldsymbol{b}_\text{rel}$ is defined as a binary variable ({0, 1}) indicating which of the two responses in a pair exhibits a stronger bias.
>
> hidden\_size: For the intermediate hidden layer size, we conducted an ablation study over the values [512, 1024, 2048]. We observed that hidden\_size=1024 offered the best trade-off between debiasing performance and minimal computational overhead.
>
> We intentionally designed $q_\psi$ to be a simple and efficient network. This ensures that the observed performance gains are attributable to our method itself, rather than to the introduction of a large number of additional parameters. We have provided the full implementation details in the supplementary material for complete transparency. The code for this network can be found in the uploaded file DIR/reward\_models/debias\_trainer\_length.py, specifically at line 230.

---

> ### Comment · Reviewer_UoY4 · 2025-11-26
> **Response to authors**
>
> Thanks for the authors' valuable time and efforts for the detailed response.
>
> I have only a question whether these supplemented results and discussions will be updated in the revised manuscript. It seems that the manuscript is not updated after submitting the response. Overall, I think this is a paper above the acceptance threshold. Currently, I maintain my rating of 6.
>
> Good luck to the authors!

---

> > ### Author Response · Authors · 2025-11-26
> >
> > Thank the reviewer UoY4 for your insightful questions and suggestions! We have incorporated them into our revised paper, and we are happy to hear that we addressed your concerns!

---

### Official Review · Reviewer_6hPr · 2025-11-01

**Soundness:** 3
**Presentation:** 3
**Contribution:** 2
**Rating:** 4
**Confidence:** 3

**Summary:**

The paper introduces a method called DIR (Debiasing via Information Optimization for Reward Models) aimed at mitigating inductive biases, such as response length, sycophancy, and format, that plague reward models in RLHF settings. It frames training as an information-theoretic problem: the method maximises the mutual information (MI) between model preferences and input response pairs, while simultaneously minimising the MI between the model’s internal representation and specified bias attributes. The debiasing objective is realised via an adversarial variational network applying the CLUB MI-estimator to ensure the learned representation is less correlated with bias attributes.

**Strengths:**

1. The paper propose a new method and tackles well-documented biases in reward models
2. the approach provides a  structure that could extend to multiple bias types.
3. The proposed DIR method shows improvements over several baselines

**Weaknesses:**

1. The paper presents itself as introducing a “novel information-theoretic framework,” but its core components are repurposed versions of existing methods. The preference loss is simply the standard Bradley-Terry ranking loss, reinterpreted post hoc as a mutual-information maximization objective. The debiasing term also relies on a conventional adversarial setup using the CLUB estimator [1], a technique already established in prior work. Although the implementation is sound and practically useful, it does not represent a genuinely new theoretical contribution.
2. The paper’s analysis of the key hyperparameter, $\lambda$, is self-contradictory. Figure 4 claims that performance peaks at $\lambda = 1$ and drops sharply at $\lambda = 10$ , calling the latter an “over-correction.” However, Table 5 shows the opposite: the $\lambda = 10$  model achieves the highest overall and “Hard” subset scores. These results directly conflict, undermining the credibility of the authors’ claims about the optimal debiasing strength and the rigor of their tuning methodology.
3. Although the results are generally strong, they are not consistently superior across benchmarks. In the Length Bias test, the proposed method’s correlation (0.468) only slightly outperforms the Skywork baseline (0.498), showing minimal real improvement. In the Sycophancy Bias test, the InfoRM baseline even outperforms the proposed method in the 20%/70% adversarial setting (86.6 vs. 85.1). These mixed outcomes weaken the claim of “most consistent and robust performance” and suggest the improvements may be context-dependent rather than universal.
4. The method’s dependence on a pre-defined and labeled bias attribute ($b$) is a practical limitation. It requires the user to already know what the bias is (e.g., length, sycophancy) and be able to label it for every data pair. This makes it useless against unknown or hard-to-quantify biases, a limitation that more "indirect" methods like InfoRM (which the paper critiques) [2] would not have.

[1] Club: A contrastive log-ratio upper bound of mutual information.
[2] Inform: Mitigating reward hacking in rlhf via information-theoretic reward modeling.

**Questions:**

1. The DIR framework's primary practical limitation is that it requires a "pre-defined bias attribute b". This means a human must identify and label the specific bias (length, format, etc.) they want to remove. How does this method fare against unknown or unlabeled biases, which are common in real-world data?

---

> ### Author Response · Authors · 2025-11-20
> **Response to Weakness-1: Novelty and Contribution; Weakness-2: Hyper-parameter Analysis;**
>
> **Response to Weakness-1: Novelty and Contribution**
>
> We thank the reviewer for your valuable feedback and fully understand the concern: we indeed do not claim to advance information theory itself or invent new MI estimators. **To the best of our knowledge, we are the first to formulate debiased reward modeling in RLHF as an explicit mutual-information optimization over the preference and bias variables themselves, and to turn this into a concrete, end‑to‑end training framework.**
>
> Concretely, our high-level objective (Eq. 8) poses debiased RM as maximizing:
>
> $$
> I\big(\mathbf{1}_{\boldsymbol{y} \succ \boldsymbol{\overline{y}}}; \boldsymbol{x}, \boldsymbol{y}, \boldsymbol{\overline{y}}\big),
> $$
>
> while explicitly minimizing:
>
> $$
> I \big(\mathbf{1}_{\boldsymbol{y} \succ \boldsymbol{\overline{y}}}; \boldsymbol{b}\big),
> $$
>
>  with respect to a specified bias attribute $\boldsymbol{b}$. This differs from IB-style approaches (i.e., InfoRM) that regularize a latent representation without explicitly modeling the dependence between the preference signal and an identified bias variable. Our formulation unifies what prior work typically treats heuristically (e.g., penalizing Pearson correlation with length, or using generic IB compression) into a single, principled target directly tied to the preference and bias random variables. Within this framework, we then (1) **make explicit the link between the standard BT loss and a BA lower bound on $I\big(\mathbf{1}_{\boldsymbol{y} \succ \boldsymbol{\overline{y}}}; \boldsymbol{x}, \boldsymbol{y}, \boldsymbol{\overline{y}}\big)$ in the RM context (to our knowledge not previously formalized in RM)**, and (2) **design a comparative MI regularizer on $\Delta \boldsymbol{h}$ and $\boldsymbol{b}_{\text{rel}}$ using CLUB that matches the pairwise structure of RM training and empirically outperforms naive concatenation-based variants (Table 6).** We will revise our manuscript as an RLHF-/RM-specific information-theoretic modeling and debiasing framework, built by explicitly combining existing MI lower and upper bounds around preference and bias.
>
> **Response to Weakness-2: Hyper-parameter Analysis**
>
> The difference in the optimal $\lambda$ lies in **a direct result of evaluating on two distinct benchmarks (RewardBench and RM-Bench) with different difficulty levels, which also reveals an important insight about the nature of debiasing.**
>
> The ablation in Figure 4 was conducted on RewardBench, which is relatively easier to fit demonstrated by several previous work [1-4]. On these tasks, a moderate debiasing strength ($\lambda=1$) proves optimal for overall performance, as it effectively reduces common biases without "over-correcting" on simpler problems. This is why we selected $\lambda=1$ as the default for our main PPO experiments, aiming for a well-balanced model.
>
> In contrast, Table 5 uses RM-Bench, which is a more challenging benchmark and specifically designed to evaluate reward models on "subtlety and style," including a particularly difficult "Hard" subset by delicate style and length control. On these complex tasks, superficial biases like length become potent vectors for reward hacking. Here, a stronger debiasing coefficient ($\lambda=10$) forces the model to ignore these misleading inductive biases and learn deeper, more substantive signals of response quality. This explains the dramatic performance leap on the "Hard" subset and its better overall score on this specific benchmark, while we also observe a similar decreased performance on the "Easy" subset.
>
> In essence, our results are complementary and reveal a critical insight: **the optimal debiasing strength should be task-dependent and correlates with task difficulty.** A moderate $\lambda$ should be best for general-purpose performance, while a strong $\lambda$ is better for creating robust models for specialized, challenging domains. Our framework's ability to tune $\lambda$ is therefore a key feature, not a contradiction, offering a practical way to tailor reward models to specific alignment goals.
>
> [1] RM-Bench: Benchmarking Reward Models of Language Models with Subtlety and Style
>
> [2] REWARDBENCH 2: Advancing Reward Model Evaluation
>
> [3] HOW TO EVALUATE REWARD MODELS FOR RLHF
>
> [4] EVALUATING ROBUSTNESS OF REWARD MODELS FOR MATHEMATICAL REASONING

---

> ### Author Response · Authors · 2025-11-20
> **Response to Weakness-3: Performance; Weakness-4 and Question-1: Pre-defined Bias**
>
> **Response to Weakness-3**
>
> We thank the reviewer for the detailed analysis and will revise the manuscript to ensure our claims are presented more rigorously. Our claim of "consistent and robust performance" stems from our method's overall effectiveness, especially its positive impact on the final alignment goal across three types of bias on most settings.
>
> Regarding Lengt: We view the Pearson correlation as the proof of successful linear decoupling at the RM-output level, while the true measure of effectiveness is reflected in the policy's resulting behavior: Table 1 shows it leads to stronger capabilities across most benchmarks and overall performance, while Figure 2 demonstrates we achieve a higher win-rate against strong baselines while being more concise. **This combination of better quality and higher efficiency is a practical achievement that goes far beyond a change in a correlation coefficient.**
>
> Regarding Sycophancy Bias: While we acknowledge the single setting where InfoRM is slightly ahead, a full analysis of Table 3 reveals a clear pattern of our method's effectiveness. We achieve higher accuracy in **8 out of 9** diverse settings for *All.* and *Adv.* metric. Crucially, our performance improves as the contamination rate increases (from $\gamma=20$%  to $\gamma=80$%). At the highest contamination level ($\gamma=80$%), where the bias signal ratio in the training dataset is strongest, our method performs best across all conditions. **This resilience under pressure, without harming performance on clean "Natural" data, is the essence of what we consider robust.**
>
>
>
> **Response to Weakness-4 and Question-1**
>
> We thank the reviewer for highlighting this crucial point. We respectfully argue that requiring a pre-defined $\boldsymbol{b}$ is a design choice. **Our goal is to achieve precise, verifiable debiasing on important alignment issues while remaining practical for real-world RLHF.**
>
> 1. Precision and reliability vs. unsupervised approaches. **Conceptually**, by explicitly specifying a bias $\boldsymbol{b}$ and minimizing $I(\Delta \boldsymbol{h}; \boldsymbol{b})$, DIR gives a clear and controllable handle on which spurious signal is suppressed and how strongly (via $\lambda$). In contrast, unsupervised ways like InfoRM compress the representation without supervision on any specific bias and may still preserve informative but spurious cues if they correlate with true preference, or conversely, compress away information that is actually important for capturing human preference. In such cases, even if the bias is partially reduced, the resulting reward model may not be effective for RLHF, which is also reflected in our results **empirically**: Table 1 shows that InfoRM does not consistently improve over the base models, while DIR yields the largest average gains. In Figure 2(b), InfoRM reduces response length to a similar level as ours, but Figure 2(a) shows noticeably lower win rates, indicating *a weaker trade-off between debiasing and performance*. By directly targeting the known bias, DIR is mainly designed to remove the spurious component while preserving preference-relevant information, thereby achieving both better direct debiasing and higher preference quality.
>
> 2. High practicality in RLHF. In many RLHF applications, various significant problematic hacking behaviors (length, formatting artifacts, sycophantic phrases, preference style) are usually known and can be perceived. For these cases, specifying $\boldsymbol{b}$ or $\boldsymbol{b}_\text{rel}$ is cheap, and DIR provides a targeted, efficient way to neutralize exactly those biases we care about.
>
> 3. Alignment with prior work. Our setting follows the mainstream paradigm in both reward-hacking work [1–5] and border NLP debiasing [6–9], where defining the attribute to be neutralized (e.g., length, gender, race) is a prerequisite for effective intervention. DIR extends this paradigm with an information-theoretic framework for reward models: once $\boldsymbol{b}$ is defined, we offer a principled way to reduce its mutual information (MI) with the preference signal.
>
> In summary, we position DIR as a practical tool for some known, high-impact biases, complementary to more exploratory unsupervised methods. We will clarify this setting in our revised related work.
>
> [1] From lists to emojis: How format bias affects model alignment
>
> [2] Beyond excess and deficiency: Adaptive length bias mitigation in reward models for rlhf.
>
> [3]  Odin: Disentangled reward mitigates hacking in rlhf
>
> [4] Beyond reward hacking: Causal rewards for large language model alignment
>
> [5] Inform: Mitigating reward hacking in rlhf via information-theoretic reward modeling
>
> [6] Examining gender and race bias in two hundred sentiment analysis systems.
>
> [7] FAIRFIL: CONTRASTIVE NEURAL DEBIASING METHOD FOR PRETRAINED TEXT ENCODERS
>
> [8] Measuring and mitigating unintended bias in text classification.
>
> [9] On Measuring Social Biases in Sentence Encoders

---

> > ### Comment · Reviewer_6hPr · 2025-11-21
> > **Response to the Authors**
> >
> > Thank you to the authors for the clarifications. I am raising my score toward acceptance and hope the manuscript will be revised accordingly. I wish the authors the best of luck!

---

> > > ### Author Response · Authors · 2025-11-22
> > >
> > > Thank the reviewer 6hPr for your raised score, and we are happy to hear that we addressed your concerns! We will include your insightful suggestions in our revised paper!

---

### Official Review · Reviewer_jfKf · 2025-11-02

**Soundness:** 2
**Presentation:** 2
**Contribution:** 3
**Rating:** 6
**Confidence:** 3

**Summary:**

The paper proposes an information-theoretical viewpoint on reward modeling and the Bradley-Terry model. Specifically, the proposed method, DIR, focuses on mitigating inductive biases, such as verbosity bias and stylistic biases, by maximizing the mutual information between preference prediction and input-response pairs. By demonstrating the performance of the reward model itself and as a preference proxy in RLHF training, the paper shows that DIR could be an effective debiasing objective for reward modeling.

**Strengths:**

1. The paper presents an intuitive yet theoretically reasonable scope of understanding reward modeling as aligning the preference distribution and preference prediction from the reward model.
2. The benchmark analysis on the biases in the reward model benchmark, RM-Bench, comes before the actual debiasing evaluation of the proposed method, which strengthens the experimental rigor of the paper.
3. Alongside the well-known length bias, the paper studies multiple types of biases and demonstrates that DIR can be an effective learning objective across different biases.

**Weaknesses:**

The main weakness of the paper is in the clarity of writing. The clarity of mathematical notations and experimental details in the paper can be improved. Other points that could either be clarified or stated as weaknesses are listed in the questions. Overall, the clarity in Sections 2 and 3 should be improved for better clarity. While there are multiple cases where the notational consistency/clarity is lacking, these are a few examples:
- Section 3.1 starts by saying that $\mathcal{L}\_\text{total}$ consists $\mathcal{L}\_\text{pref}$ and $\mathcal{L}\_\text{debias}$, while Equation (12) uses $\mathcal{L}\_\text{reward}$ instead.
- $\mathcal{L}_\text{debias}$ is not explicitly defined in the paper and appears in Equation (12).

These inconsistencies and missing definitions prevent a clear understanding of the paper, even though the paper's theoretical soundness should be highlighted as its strength.

**Questions:**

- Can DIR be expanded to the implicit reward models like direct alignment algorithms?
- On the official RM-Bench leaderboard, “Skywork-Reward-Llama-3.1-8B-v0.2” (“BT” in Table 5) has mostly higher numbers compared to the results stated in Table 5. For example, “Hard” score for Skywork-Reward-Llama-3.1-8B-v0.2 is 52.6 and 69.3 for “Chat”, while the reported scores in the paper are 42.76 and 64.69, respectively. Given that the numbers on the leaderboard could change the trend in Table 5 (e.g., the overall score of Skywork-Reward-Llama-3.1-8B-v0.2 on the leaderboard is higher than “Ours-10.0”), this part needs clarification.
- On the RM-Bench scores, why does the “Ours” model experience a notable drop in the “Easy” accuracy? By debiasing, is the model experiencing a trade-off in easy stylistic differentiation?

---

> ### Author Response · Authors · 2025-11-20
> **Response to Weakness: Inconsistent Notations; Question-2: Performance; Question-3: Trade-Off**
>
> **Response to Weakness: Inconsistent Notations**
>
> We thank the reviewer for carefully pointing out these issues, and will make sure the revised version presents the math and experimental details in a much clearer and more consistent way by following your suggestion!
>
> **Response to Question-2: Performance**
>
> First, we apologize for the confusion: the “BT” model in Table 5 is our own re-implementation of a standard BT reward model trained on Skywork-Preference-80K-v0.2, not the public “Skywork-Reward-Llama-3.1-8B-v0.2” checkpoint. Since the official training code and full training framework of Skywork-Reward is not public, we trained the baseline within our own codebase (following the hyperparameters reported in [1]) to isolate the effect of our debiasing method and ensure a fair, controlled comparison.
>
> Second, our re-implementation uses only one training epoch, motivated by prior work [2–4] showing that reward models easily overfit with multiple epochs. The official Skywork-Reward-Llama-3.1-8B-v0.2 model configuration file indicates that it was trained for more than one epoch. The difference in training regime, hardware environment, together with implementation details, leads to lower absolute RM-Bench scores for our BT baseline compared to the leaderboard entry.
>
> To address the reviewer’s concern, we additionally evaluated the official “Skywork-Reward-Llama-3.1-8B-v0.2” checkpoint in our own environment on RM-Bench, as shown below. While the absolute numbers are lower than those on the public leaderboard (e.g., Hard: 46.38\% vs. 52.6\%, Chat: 68.04\% vs. 69.3\%), the relative trend remains: our Ours-10.0 model still achieves a higher total score (70.18 vs. 69.66) and substantially better Hard performance (64.41 vs. 46.38). We will add this comparison table and explicitly clarify in the revised paper.
>
>
> | Method      | Chat      | Math      | Code      | Safety    | Hard      | Normal    | Easy        | Total      |
> |------------|-----------|-----------|-----------|-----------|-----------|-----------|-------------|------------|
> | BT         | 64.69     | 61.21     | 51.41     | 95.11     | 42.76     | 72.30     | **89.24**   | 68.10      |
> | SKRM-v0.2  | 68.04     | 61.18     | **52.88** | **96.52** | 46.38     | **73.77** | 88.82       | _69.66_    |
> | Ours-1.0   | _68.91_   | **61.81** | 51.56     | _95.13_   | _47.88_   | _73.59_   | 88.93     | 69.35      |
> | Ours-10.0  | **71.23** | _61.59_   | _52.73_   | 94.91     | **64.41** | 71.29     | 74.85       | **70.18**  |
>
> [1] Skywork-Reward: Bag of Tricks for Reward Modeling in LLMs
>
> [2] Training language models to follow instructions with human feedback
>
> [3] Iterative data smoothing: Mitigating reward overfitting and overoptimization in rlhf
>
> [4] Helpsteer2-preference: Complementing ratings with preferences
>
>
> **Response to Question-3: Trade-Off**
>
> We agree that this is a trade-off. As the RM-Bench paper notes, many “Easy” examples can be solved by simple stylistic heuristics (e.g., length or format correlating with quality), whereas “Hard” examples are specifically constructed so that these shortcuts become misleading. Our debiasing objective explicitly penalizes reliance on such spurious cues, which improves robustness and yields large gains on “Hard” (e.g., +21.65 points for Ours-10.0 over BT) and overall scores, but naturally makes the model less willing to exploit those shortcuts on “Easy”, leading to a drop there.
>
> **Crucially, this trade-off is also controllable via the debiasing coefficient $\lambda$.** As shown by comparing Ours-1.0 and Ours-10.0 in Table 5 (and further detailed in our $\lambda$ ablation in Appendix C.2), increasing $\lambda$ strengthens debiasing, improves Hard and overall performance, but comes with a larger decrease on Easy. We can therefore choose $\lambda$ according to the tolerance for such stylistic shortcuts: smaller $\lambda$ if preserving easy-heuristic performance is important, or larger $\lambda$ if robustness on challenging, bias-sensitive scenarios is prioritized.

---

> ### Author Response · Authors · 2025-11-20
> **Response to Question-1: Combination with DPO**
>
> We thank the reviewer for this excellent question. Conceptually, DIR operates at the reward modeling stage and should not modify the DPO objective: **DPO still optimizes the standard log-sigmoid preference loss with the same $\beta$, and our method only modifies preference signals by making them less correlated with inductive bias**. Empirically, we conducted the corresponding experiments, which show that **our method can also improve DPO's performance with controlled length.**
>
> Specifically, we adopt ms-swift framework [1] with its default DPO training configuration on Human-Like-DPO-Dataset[2], based on both OpenRLHF-Llama-3-8b-SFT and Meta-Llama3.1-8B-Instruct models, where DPO $\beta=0.1$, debias factor $\lambda=1$. We train 1 epoch and evaluate the performance on the final checkpoint. Human-Like-DPO-Dataset is created to fine-tune LLMs toward generating more human-like responses, which includes 10,884 samples across 256 topics, containing technology, daily Life, science, history and arts. We evaluate the performance on ArenaHard-v0.1 and several popular benchmarks. For baselines, we also compare with length-controlled DPO method [3], which is designed to explicitly avoid the policy model from preferring the longer response during DPO training. As shown in the Table 1 below, we find that **our method can lead to a final policy that is both higher quality and more token-efficient:**
>
> Table 1. ArenaHard Win Rate (\%)
>
> | Win Rate (%)        | OpenRLHF-Llama-3-8B-SFT |                  | Meta-Llama3.1-8B-Instruct |                  |
> |---------------------|-------------------------|------------------|---------------------------|------------------|
> | vs Base             | ArenaHard-v0.1          | Length           | ArenaHard-v0.1            | Length           |
> | DPO                 | 38.63                   | 436.55           | 44.06                     | 700.87           |
> | DPO + LengthControl | 40.96                   | 407.23           | 46.57                     | 691.43           |
> | DPO + OURS          | **45.27**                   | **404.61**           | **49.09**                     | **684.67**           |
>
> Results on ArenaHard indicate that our method leads to a final policy with both a better win-rate and more effective length control, effectively boosting vanilla DPO and also outperforming a specialized Length Controlled DPO (DPO-LC) variant [3].
>
> Results on Table 2 indicate that **our method also leads to a final policy with performance gains, especially for SFT model, which has not undergone a preference alignment. **
>
> Table 1. Benchmark Performance
>
> | Benchmark                         | Meta-Llama3.1-8B-Instruct DPO | DPO-LC | DPO+OURS | OpenRLHF-Llama3-8B-SFT DPO | DPO-LC | DPO+OURS |
> |-----------------------------------|-------------------------------|--------|----------|-----------------------------|--------|----------|
> | GSM8K_acc-4shots                  | 82.11                         | 81.43  | 82.56    | 75.89                       | 76.35  | 77.26    |
> | Hellaswag_acc                     | 74.84                         | 75.17  | 75.10    | 66.56                       | 66.41  | 73.91    |
> | IFeval_acc                        | 73.57                         | 74.12  | 74.68    | 38.26                       | 35.86  | 41.59    |
> | MMLU_acc                          | 71.36                         | 71.58  | 71.54    | 48.92                       | 48.92  | 57.01    |
> | ProcessBench_acc                  | 26.75                         | 26.70  | 27.60    | 4.28                        | 4.95   | 6.93     |
> | Race_acc-3shots                   | 69.98                         | 70.14  | 70.29    | 79.27                       | 78.68  | 79.97    |
> | BBH_acc                           | 67.22                         | 66.81  | 66.99    | 59.73                       | 60.36  | 62.34    |
> | Humaneval_pass@1                  | 62.80                         | 65.24  | 69.51    | 59.15                       | 60.37  | 59.15    |
> | TriviaQA_acc-5shots               | 55.12                         | 55.15  | 55.29    | 47.45                       | 47.69  | 48.93    |
> | Avg. Performance                  | 64.86                         | 65.04  | 65.95    | 53.28                       | 53.29  | 60.12    |
> | Δ                                 | -                             | 0.18   | **1.09**     | -                           | 0.01   | **6.84**     |
>
> In summary, experiments in Table 1 and Table 2 suggest that** debiased reward signals from DIR interact smoothly with DPO and effectively remove spurious gradients induced by length bias**. We will include a more detailed analysis of these DPO experiments in our revised paper and update our uploaded codebase to ensure full reproducibility.
>
> [1] https://github.com/modelscope/ms-swift
>
> [2] https://huggingface.co/datasets/HumanLLMs/Human-Like-DPO-Dataset
>
> [3] Disentangling Length from Quality in Direct Preference Optimization

---

### Official Review · Reviewer_cNct · 2025-11-07

**Soundness:** 3
**Presentation:** 3
**Contribution:** 3
**Rating:** 6
**Confidence:** 4

**Summary:**

This paper proposes DIR, an information-theoretic framework for debiasing reward models in RLHF. The method maximizes MI between predictions and true preferences while minimizing MI between internal representations and predefined bias attributes. Using data processing inequality and variational bounds (BA and CLUB), the theoretical objective becomes a tractable loss. Experiments on length, sycophancy, and format biases show improvements in both RM metrics and downstream RLHF performance.

**Strengths:**

1. The explicit use of data processing inequality to justify representation-level debiasing, combined with dual variational bounds (BA for information retention, CLUB for bias suppression), provides an elegant and principled solution.

2. Experiments cover three diverse bias types with end-to-end assessment (RM performance + downstream PPO policies). Strong ablations on representation choice (Table 6) and hyperparameter $\lambda$ (Figure 4) validate design decisions.

3. Zero inference overhead and moderate training cost (~33% increase). Demonstrated versatility across bias types suggests broad applicability.

**Weaknesses:**

1. Method requires knowing bias types *a priori* and labeling $b_{\mathrm{rel}}$ for every pair. No mechanism for unsupervised bias discovery limits real-world applicability.
2. Experiments isolate single biases. Real datasets likely contain concurrent biases (e.g., lengthy + sycophantic responses). Unclear how to extend DIR—multiple debiasing terms with separate $\lambda$ values? Potential optimization conflicts?
3. Sycophancy evaluation uses fixed prefix injection (“*Yes, you are right.*”). Real biases are more subtle and contextually integrated, raising generalization concerns.
4. Unexplored Representation Alternatives: Exclusively uses final hidden state without comparing alternatives (e.g., mean-pooling). Global representations may better capture stylistic/format biases.

**Questions:**

1. How would DIR handle concurrent biases? Would you use $\mathcal{L}\_{\text{reward}} + \sum\_i \lambda\_i \mathcal{L}\_{\text{debias}}^{i}$? What optimization challenges arise from negative interactions between debiasing signals?
2. Have you studied the architecture sensitivity of $q_{\psi}$? Could an overly powerful $q_{\psi}$ discard legitimate preference-informative correlations along with spurious biases?
3. How do debiased RMs interact with direct alignment methods like DPO? Could altered reward landscapes complicate implicit differentiation?

---

> ### Author Response · Authors · 2025-11-20
> ****Response to Weakness-1: Pre-defined Bias****
>
> We thank Reviewer cNct for highlighting this crucial point. We respectfully argue that requiring a pre-defined bias attribute $\boldsymbol{b}$ is a design choice, rather than an inherent limitation of DIR. Our goal is to achieve precise, verifiable debiasing on important alignment issues while remaining practical for real-world RLHF.
>
> 1. Precision and reliability vs. unsupervised approaches. **Conceptually**, by explicitly specifying a bias $\boldsymbol{b}$ and minimizing $I(\Delta \boldsymbol{h}; \boldsymbol{b})$, DIR gives a clear and auditable handle on which spurious signal is suppressed and how strongly (via $\lambda$). In contrast, unsupervised methods like InfoRM compress the representation without supervision on any specific bias and may still preserve informative but spurious cues if they correlate with true preference, or conversely, compress away information that is actually important for capturing human quality judgments. In such cases, even if the bias is partially reduced, the resulting reward model may not be effective for RLHF, which is also reflected in our results **empirically**: Table 1 shows that InfoRM does not consistently improve over the base models on either backbone, while DIR yields the largest average gains. In Figure 2(b), InfoRM reduces response length to a similar level as ours, but Figure 2(a) shows noticeably lower win rates, indicating *a weaker trade-off between debiasing and performance*. By directly targeting the known bias, DIR is mainly designed to remove the spurious component while preserving preference-relevant information, thereby achieving both better direct debiasing and higher preference quality.
>
> 2. High practicality in RLHF. In many RLHF applications, various significant problematic reward-hacking behaviors (length, formatting artifacts, sycophantic phrases, preference style) are usually known and can be perceived, thus trivial to label or compute. For these cases, specifying $\boldsymbol{b}$ or $\boldsymbol{b}_\text{rel}$ is cheap, and DIR provides a targeted, efficient way to neutralize exactly those biases we care about.
>
> 3. Alignment with prior work. Our setting follows the mainstream paradigm in both reward-hacking work [1–5] and border NLP debiasing [6–9], where defining the attribute to be neutralized (e.g., length, format, gender, race) is a prerequisite for effective intervention. DIR extends this paradigm with an information-theoretic framework for reward models: once $\boldsymbol{b}$ is defined, we offer a principled way to reduce its mutual information (MI) with the preference signal.
>
> In summary, we position DIR as a practical tool for some known, high-impact biases, complementary to more exploratory unsupervised methods. We will clarify this setting in our revised related work.
>
> [1] From lists to emojis: How format bias affects model alignment
>
> [2] Beyond excess and deficiency: Adaptive length bias mitigation in reward models for rlhf.
>
> [3]  Odin: Disentangled reward mitigates hacking in rlhf
>
> [4] Beyond reward hacking: Causal rewards for large language model alignment
>
> [5] Inform: Mitigating reward hacking in rlhf via information-theoretic reward modeling
>
> [6] Examining gender and race bias in two hundred sentiment analysis systems.
>
> [7] FAIRFIL: CONTRASTIVE NEURAL DEBIASING METHOD FOR PRETRAINED TEXT ENCODERS
>
> [8] Measuring and mitigating unintended bias in text classification.
>
> [9] On Measuring Social Biases in Sentence Encoders

---

> ### Author Response · Authors · 2025-11-20
> **Response to Weakness-2 and Question-1: Concurrent Multi-Biases**
>
> We thank the reviewer for this insightful question and fully agree that real datasets often exhibit multiple concurrent biases. We conducted a new multi-bias experiment where DIR simultaneously debiases both length and sycophantic responses. Overall, we find that **the multi-bias DIR reduces both length and sycophancy bias compared to the BT baseline, indicating that multiple debiasing terms can be combined without significant destructive optimization conflicts.**
>
> Concretely, we extend DIR to these two explicit biases by introducing two independent Mutual-Information regularizers, each with its own $q_\psi$ head. For each preference pair, we construct the corresponding two bias labels, and optimize:
>
> $$
> \mathcal{L} = \mathcal{L}_{\text{reward}} + \lambda_{\text{len}} * \mathcal{L}_{\text{debias_len}} + \lambda_{\text{syco}} * \mathcal{L}_{\text{debias_syco}}
> $$
>
> where $\lambda_{\text{len}} = \lambda_{\text{syco}}=1$. We follow the Table 3 setting, training Meta-Llama-3.1-8B-Instruct on HelpSteer3 dataset under two sycophancy contamination configurations ($\gamma = 0.4 / 0.8, \alpha = 0.7$), where a larger $\gamma$ indicates a more challenging setting. All models (BT, length-only DIR, and length+syco DIR) are trained for 1 epoch on the same data, and we report results using the final checkpoint for fairness and convenience.
>
> As shown in Table 1, on RM-Bench, the joint model (OURS-Len-Syco) achieves the best overall performance and the largest gains on the hardest subset (e.g., at $\gamma=0.4$, Total: $67.25 \rightarrow 69.96$, Hard: $39.88 \rightarrow 46.85$ vs. BT), *while still clearly reducing the Pearson correlation with length relative to BT, confirming that length bias is mitigated even when sycophancy is also debiased*. We also observe that the length-only model (OURS-Len) attains the lowest length–reward Pearson coefficient, but somewhat surprisingly, the joint length+sycophancy debiasing (OURS-Len-Syco) yields the best overall RM-Bench performance, suggesting that *debiasing multiple biases together may help lead to a more balanced and effective reward model.*
>
> Table 1. RM-Bench performance on the length bias
>
> |  $\gamma=40$%, $\alpha=70$%     | Chat  | Math  | Code  | Safety | Hard  | Normal | Easy  | Total | Pearson Coefficient |
> |-------------------|-------|-------|-------|--------|-------|--------|-------|-------|---------------------|
> | BT                | 66.58 | 64.17 | 53.70 | 84.53  | 39.88 | 72.82  | **89.04** | 67.25 | 0.4807              |
> | OURS-Len          | 66.93 | 64.59 | 53.12 | **89.14**  | 44.25 | 72.43  | 88.65 | 68.44 | **0.4235**              |
> | OURS-Len-Syco     | **70.80** | **65.07** | **55.26** | 88.71  | **46.85** | **75.08**  | 87.96 | **69.96** | 0.4446              |
>
> |  $\gamma=80$%, $\alpha=70$%      | Chat  | Math  | Code  | Safety | Hard  | Normal | Easy  | Total | Pearson Coefficient |
> |-------------------|-------|-------|-------|--------|-------|--------|-------|-------|---------------------|
> | BT                | 65.37 | 63.71 | 53.12 | 80.85  | 37.58 | 70.96  | **88.75** | 65.76 | 0.4666              |
> | OURS-Len          | **68.91** | 64.25 | 53.75 | 87.23  | 44.78 | **73.09**  | 87.72 | 68.53 | **0.4081**              |
> | OURS-Len-Syco     | 68.39 | **64.92** | **54.04** | **88.06**  | **45.15** | 72.92  | 88.50 | **68.85** | 0.4235              |
>
>
> As shown in Table 2, on sycophancy stress tests, debiasing only sycophancy (OURS-Syco) gives the strongest syco robustness, as expected, but the joint model (OURS-Syco-Len) still *substantially outperforms BT on all sycophancy metrics (All/Nat/Adv) across both $\gamma$ settings, while additionally reducing length bias.*
>
> Table 2. Reward model accuracy (%) on the sycophancy bias
>
> | $\gamma$   | $\alpha$    | All. BT | All. OURS-Syco | All. OURS-Syco-Len | Nat. BT | Nat. OURS-Syco | Nat. OURS-Syco-Len | Adv. BT | Adv. OURS-Syco | Adv. OURS-Syco-Len |
> |------|------|---------|----------------|---------------------|--------|----------------|---------------------|---------|----------------|---------------------|
> | 40%  | 70%  | 83.6    | 88.0           | 85.6                | 84.4   | 87.4           | 86.4                | 82.6    | 88.6           | 84.4                |
> | 80%  | 70%  | 81.2    | 86.2           | 85.9                | 86.4   | 87.2           | 86.6                | 79.7    | 86.2           | 85.1                |
>
>
> In summary, a second debiasing term leads to a controlled trade-off, not conflicting gradients: both biases are improved over BT, and overall RM quality remains strong. We will complete the experiments and analysis, add these results in our revised paper.

---

> ### Author Response · Authors · 2025-11-20
> **Response to Weakness-3: Fixed Prefix Injection; Weakness-4: Representation Alternatives; Question-2: Architecture Sensitivity**
>
> **Response to Weakeness-3: Fixed Prefix Injection:**
>
> We thank the reviewer for this insightful point and agree that real-world sycophancy is often more subtle. Our use of a controlled prefix injection is an experimental choice, consistent with prior work [1–2], to obtain a clean and reproducible testbed. Importantly, our contamination scheme is not a trivial “detect the string” setting: for each contamination setting ($\lambda$, $\alpha$), the same prefix (“Yes, you are right.”) is injected into the preferred response only with probability $\alpha$ and into the rejected response with probability $1-\alpha$. Thus, the prefix is a strong but unreliable cue, and a reward model cannot simply memorize it; it must learn to the this spurious signal to perform well across both natural and adversarial splits. The consistent gains of DIR over BT and InfoRM under multiple combinations suggest robustness to such strong yet non-deterministic bias rather than to a single fixed pattern. Due to time limits, we have to say that it's hard for us to add more realistic experiments during the rebuttal, but we will clarify the generality in the revision and add an experiment on extending more nuanced sycophancy bias setting.
>
> [1] Towards understanding sycophancy in language models.
>
> [2] Beyond reward hacking: Causal rewards for large language model alignment
>
> **Response to Weakness-4: Representation Alternatives:**
>
> Our choice to use the final hidden state (i.e., the representation of the [EOS] token) is rooted in the established practice of current reward modeling implementation [1-3]. Autoregressive models are pre-trained to use the final token's hidden state as a comprehensive summary of the entire preceding sequence. Through the self-attention mechanism, the final state has aggregated information from all previous tokens, which has been a global representation with respect to the current input, including both query and response. Therefore, during reward model training, the final representation should be naturally optimized to encode all necessary information for making a sequence-level preference judgment. Besides, in our setting, we further operate on the difference between two such representations ($\Delta h$), which focuses the debiasing signal precisely on the features that distinguish the two responses (including their style and format), rather than on token-level details.
>
> [1] Let’s Verify Step by Step.
>
> [2] Line545, \url{https://github.com/huggingface/trl/blob/main/trl/trainer/reward_trainer.py}
>
> [3] Line310, \url{https://github.com/OpenRLHF/OpenRLHF/blob/main/openrlhf/trainer/rm_trainer.py}
>
> **Response to Question-2: Architecture Sensitivity:**
>
> Yes, we studied architectural sensitivity. Our variational network $q_\psi$ for estimating the mutual information is implemented as a lightweight two-layer Multi-Layer Perceptron, where we performed an ablation on the hidden_size of $q_\psi$ over the values [512, 1024, 2048] and found that 1024 provided the best balance of performance and efficiency. Our design philosophy is to keep $q_\psi$ intentionally lightweight, which could ensure the success of debias stems from its principled objective, rather than from adding excessive parameters. Regarding the risk that an overly powerful $q_\psi$ might remove legitimate preference-informative correlations together with spurious ones, we note that (1) $q_\psi$ only operates on $\Delta \boldsymbol{h}$ and the bias label $\boldsymbol{b}\_\text{rel}$, so it can only push the representation to become invariant to the specified bias attribute, not to arbitrary aspects of the preference signal; and (2) the strength of this pressure is explicitly controlled by the coefficient $\lambda$. As shown in our $\lambda$ ablation (Appendix C.2), when a too large $\lambda$ directly reflects the “over-debiasing” effect. In our main experiments, we choose $\lambda$ in the regime where RewardBench/RM-Bench performance is maintained or improved while bias metrics are reduced, indicating that $q_\psi$ is not discarding useful preference information in practice. We will clarify these design choices and observations in the revised version.

---

> ### Author Response · Authors · 2025-11-20
> **Response to Question-3: Combination with DPO**
>
> We thank the reviewer for this excellent question. Conceptually, DIR operates at the reward modeling stage and should not modify the DPO objective: **DPO still optimizes the standard log-sigmoid preference loss with the same $\beta$, and our method only modifies preference signals by making them less correlated with inductive bias**. Empirically, we conducted the corresponding experiments, which show that **our method can also improve DPO's performance with controlled length.**
>
> Specifically, we adopt ms-swift framework [1] with its default DPO training configuration on Human-Like-DPO-Dataset[2], based on both OpenRLHF-Llama-3-8b-SFT and Meta-Llama3.1-8B-Instruct models, where DPO $\beta=0.1$, debias factor $\lambda=1$. We train 1 epoch and evaluate the performance on the final checkpoint. Human-Like-DPO-Dataset is created to fine-tune LLMs toward generating more human-like responses, which includes 10,884 samples across 256 topics, containing technology, daily Life, science, history and arts. We evaluate the performance on ArenaHard-v0.1 and several popular benchmarks. For baselines, we also compare with length-controlled DPO method [3], which is designed to explicitly avoid the policy model from preferring the longer response during DPO training. As shown in the Table 1 below, we find that **our method can lead to a final policy that is both higher quality and more token-efficient:**
>
> Table 1. ArenaHard Win Rate (\%)
>
> | Win Rate (%)        | OpenRLHF-Llama-3-8B-SFT |                  | Meta-Llama3.1-8B-Instruct |                  |
> |---------------------|-------------------------|------------------|---------------------------|------------------|
> | vs Base             | ArenaHard-v0.1          | Length           | ArenaHard-v0.1            | Length           |
> | DPO                 | 38.63                   | 436.55           | 44.06                     | 700.87           |
> | DPO + LengthControl | 40.96                   | 407.23           | 46.57                     | 691.43           |
> | DPO + OURS          | **45.27**                   | **404.61**           | **49.09**                     | **684.67**           |
>
> Results on ArenaHard indicate that our method leads to a final policy with both a better win-rate and more effective length control, effectively boosting vanilla DPO and also outperforming a specialized Length Controlled DPO (DPO-LC) variant [3].
>
> Results on Table 2 indicate that **our method also leads to a final policy with performance gains, especially for SFT model, which has not undergone a preference alignment. **
>
> Table 1. Benchmark Performance
>
> | Benchmark                         | Meta-Llama3.1-8B-Instruct DPO | DPO-LC | DPO+OURS | OpenRLHF-Llama3-8B-SFT DPO | DPO-LC | DPO+OURS |
> |-----------------------------------|-------------------------------|--------|----------|-----------------------------|--------|----------|
> | GSM8K_acc-4shots                  | 82.11                         | 81.43  | 82.56    | 75.89                       | 76.35  | 77.26    |
> | Hellaswag_acc                     | 74.84                         | 75.17  | 75.10    | 66.56                       | 66.41  | 73.91    |
> | IFeval_acc                        | 73.57                         | 74.12  | 74.68    | 38.26                       | 35.86  | 41.59    |
> | MMLU_acc                          | 71.36                         | 71.58  | 71.54    | 48.92                       | 48.92  | 57.01    |
> | ProcessBench_acc                  | 26.75                         | 26.70  | 27.60    | 4.28                        | 4.95   | 6.93     |
> | Race_acc-3shots                   | 69.98                         | 70.14  | 70.29    | 79.27                       | 78.68  | 79.97    |
> | BBH_acc                           | 67.22                         | 66.81  | 66.99    | 59.73                       | 60.36  | 62.34    |
> | Humaneval_pass@1                  | 62.80                         | 65.24  | 69.51    | 59.15                       | 60.37  | 59.15    |
> | TriviaQA_acc-5shots               | 55.12                         | 55.15  | 55.29    | 47.45                       | 47.69  | 48.93    |
> | Avg. Performance                  | 64.86                         | 65.04  | 65.95    | 53.28                       | 53.29  | 60.12    |
> | Δ                                 | -                             | 0.18   | **1.09**     | -                           | 0.01   | **6.84**     |
>
> In summary, experiments in Table 1 and Table 2 suggest that** debiased reward signals from DIR interact smoothly with DPO and effectively remove spurious gradients induced by length bias**. We will include a more detailed analysis of these DPO experiments in our revised paper and update our uploaded codebase to ensure full reproducibility.
>
> [1] https://github.com/modelscope/ms-swift
>
> [2] https://huggingface.co/datasets/HumanLLMs/Human-Like-DPO-Dataset
>
> [3] Disentangling Length from Quality in Direct Preference Optimization

---

### Author Response · Authors · 2025-11-28

We would like to express our sincere gratitude to reviewers cNct, jfKf, 6hPr, UoY4, ACs, SACs and PCs for the time and effort they devoted to assessing our work. Their insightful comments, constructive suggestions, and detailed questions have been invaluable in refining and clarifying our manuscript. We regard the review process as a collaborative endeavor and greatly appreciate the reviewers’ thoughtful engagement with both the overarching motivation and the technical intricacies of our approach.

Across the reviews, we are glad to receive several consistent points of positive feedback:

**1. Principled and novel information-theoretic framework:** Reviewer cNct: ''...provides an elegant and principled solution.''; Reviewer jfKf: ''...an intuitive yet theoretically reasonable...''; Reviewer 6hPr : ''...a new method and tackles well-documented biases...'' and ''...a structure that could extend to multiple bias types....''.

**2. Strong and comprehensive experimental evaluation:** Reviewer cNct: ''...cover three diverse bias types with end-to-end assessment.'' and ''...Strong ablations...''; Reviewer jfKf: ''...strengthens the experimental rigor of the paper...''; Reviewer 6hPr : ''...shows improvements over several baselines...''; Reviewer UoY4 : ''...results are solid, covering three types of biases...''.

**3. Practicality and relevance to RLHF.** Reviewer cNct: ''...zero inference overhead and moderate training cost (~33% increase).'' and ''...broad applicability...''; Reviewer jfKf: ''...an effective learning objective across different biases...''; Reviewer UoY4 : ''...is very important for RLHF to improve alignment performance of LLMs....''.

**Overall assessment:** We are encouraged that Reviewer cNct, jfKf, and UoY4 initially give an acceptance rate of 6. During the discussion, Reviewer 6hPr highlighted that we addressed his concerns and then raised the score from 4 to 6. In addition, Reviewer UoY4 confirmed that this is a paper above the acceptance threshold with a maintained rate of 6.

In summary, the reviewers’ constructive suggestions helped us substantially improve our work. In direct response to their feedback, we have provided point-to-point responses and incorporated these meaningful suggestions into our revised paper:

1. Clearification of pre-defined bias on Page 8, Line 397.

2. New experiments in a concurrent multi-bias setting on Page 10, Line 514 and Page 24 Line 1260.

3. New results on additional benchmarks on Page 8, Line 403 and Page 20, Line 1065.

4. New exploration about the combination with DPO on Page 8, Line 427.

5. Improved writing, figures, and implementation details.

We believe these changes have strengthened the paper, and we appreciate the reviewers’ guidance in arriving at a clearer, more rigorous, and more thoroughly justified presentation.

Once again, we thank all reviewers for their encouragement and help in improving the manuscript.

---

### Meta-Review · Area_Chair_pkLi · 2026-01-03

**Summary:**

The submission initially received relatively positive reviews. The main concerns can be summarized as follows:
1. The proposed method requires knowing the bias type ahead, which limits its generalization in real-world scenarios
2. Real datasets may contain concurrent biases, how to extend the proposed method into multiple debiasing terms would be another question.
3. The presentation needs to be further improved, such as using an extra figure to illustrate the methodology.
4. More evaluation benchmarks are expected, and extra explanations about some results are necessary.

In the rebuttal, all above mentioned concerns have been well-addressed. Thus, I recommend Accept.

**Reviewer Concerns:**

All concerns have been addressed.

**Reviewer Scores:**

I think the Reviwer 6hPr will increase the rating from negative to positive, and all other positive reviewers will maintain his/her original positive ratings.

---

### Decision · Program_Chairs · 2026-01-26

Accept (Poster)